# Non-homologous end joining shapes the genomic rearrangement landscape of chromothripsis from mitotic errors

Qing Hu [1], Jose Espejo Valle-Inclán [2], Rashmi Dahiya[1], Alison Guyer[1,6], Alice Mazzagatti [1], Elizabeth G. Maurais[1], Justin L. Engel [1], Huiming Lu[3], Anthony J. Davis [3,4], Isidro Cortés-Ciriano [2] & Peter Ly [1,4,5] ✉

Mitotic errors generate micronuclei entrapping mis-segregated chromosomes, which are susceptible to catastrophic fragmentation through chromothripsis. The reassembly of fragmented chromosomes by error-prone DNA double-strand break (DSB) repair generates diverse genomic rearrangements associated with human diseases. How specific repair pathways recognize and process these lesions remains poorly understood. Here we use CRISPR/Cas9 to systematically inactivate distinct DSB repair pathways and interrogate the rearrangement landscape of fragmented chromosomes. Deletion of canonical non-homologous end joining (NHEJ) components substantially reduces complex rearrangements and shifts the rearrangement landscape toward simple alterations without the characteristic patterns of chromothripsis. Following reincorporation into the nucleus, fragmented chromosomes localize within sub-nuclear micronuclei bodies (MN bodies) and undergo ligation by NHEJ within a single cell cycle. In the absence of NHEJ, chromosome fragments are rarely engaged by alternative end-joining or recombination-based mechanisms, resulting in delayed repair kinetics, persistent 53BP1-labeled MN bodies, and cell cycle arrest. Thus, we provide evidence supporting NHEJ as the exclusive DSB repair pathway generating complex rearrangements from mitotic errors.

Chromosome segregation during mitosis must be accurately executed to maintain genome stability. Mitotic errors can result in the formation of aberrant nuclear structures called micronuclei that entrap mis-segregated chromosomes or chromosome arms. The irreversible rupture of the micronuclear envelope[1,2] triggers the loss of nucleocytoplasmic compartmentalization and the acquisition of DNA double-strand breaks (DSBs)[1,3–6]. Extensive DSBs can promote the catastrophic fragmentation of the micronucleated chromosome during the subsequent mitosis[4,7], a process termed chromothripsis[8]. Fragmented chromosomes remain spatially clustered during mitosis and reincorporate into the nucleus of one or both daughter cell(s), manifesting as sub-nuclear territories known as "micronuclei bodies" (MN bodies) during interphase that engage the DNA damage response (DDR)[4,9–12].

Chromothripsis frequently drives complex and localized genomic rearrangements owing to the error-prone reassembly of the fragmented chromosome[13]. These rearrangements are common across

[1]Department of Pathology, University of Texas Southwestern Medical Center, Dallas, TX, USA. [2]European Molecular Biology Laboratory, European Bioinformatics Institute, Wellcome Genome Campus, Hinxton, UK. [3]Department of Radiation Oncology, University of Texas Southwestern Medical Center, Dallas, TX, USA. [4]Harold C. Simmons Comprehensive Cancer Center, University of Texas Southwestern Medical Center, Dallas, TX, USA. [5]Department of Cell Biology, University of Texas Southwestern Medical Center, Dallas, TX, USA. [6]Present address: Department of Computational and Systems Biology, University of Pittsburgh, Pittsburgh, PA, USA. ✉e-mail: peter.ly@utsouthwestern.edu

diverse cancer types[8,14,15] and can be characterized by seemingly random structural variants that are clustered along one or a few chromosome(s)[16]. In addition to complex rearrangements, a diverse spectrum of chromosomal abnormalities can be generated from micronuclei formation, including simple arm-level deletions, insertions, and translocations[5,17]. Based on the sequence features at rearrangement breakpoint junctions, several DSB repair pathways have been predicted to underlie the formation of complex rearrangements following chromothripsis[5,14,17].

Multiple DSB repair pathways are operative in mammalian cells to process detrimental DNA lesions. The canonical non-homologous end joining (NHEJ) pathway is active throughout the cell cycle and directly ligates two DSB ends through the recruitment of Ku70/80 and activity of DNA-PKcs, XLF, and DNA ligase 4 (LIG4)–XRCC4[18]. Alternative end joining (alt-EJ) occurs independently of core NHEJ factors, with DNA polymerase theta (Polθ)-dependent microhomology-mediated end joining likely accounting for most alt-EJ events[19]. Whereas NHEJ ligates DSBs without homology, alt-EJ relies on the resection of DSB ends to expose short stretches of homologous sequence – known as microhomology – at the repair junction. Both homologous recombination (HR) and single-strand annealing (SSA) require more extensive DNA end resection to generate extended 3′ single-strand DNA (ssDNA) tails. HR is most active following genome duplication in S-phase, a period when ssDNA tails can invade a homologous DNA sequence (e.g., a sister chromatid) and use it as a template for synthesis to repair the DSB, which involves BRCA1, BRCA2, RAD51, and RAD54[20]. SSA requires the annealing of homologous repeats to form the synapsis intermediate before ligation, a process that is mediated by RAD52[21]. The DSB repair pathways described can be mutagenic when multiple DSBs are present and/or incorrect sequences are used for recombination[22–26].

In cancer genomes and germline disorders with chromothripsis, the majority of rearrangement breakpoints harbor blunt-ended junctions without homology; however, microhomology signatures – which can be loosely defined to include as little as one nucleotide of homology – have also been reported[8,14,15,27–31]. Although similar observations have been described in experimental models of chromothripsis[5,17,32], the extent in which specific DSB repair pathways contribute to reassembling fragmented chromosomes from micronuclei has not been systematically characterized. We previously showed that depletion of DNA-PKcs and LIG4, two important components of NHEJ, was sufficient to reduce, but not completely abrogate, the formation of rearrangements from micronuclei[7,17]. This approach was limited by a partial reduction of key genes by RNA interference, including DNA repair enzymes whose activity may remain functional at low levels. Additionally, chromothripsis from DSB repair-deficient murine tumors[33] and human cells escaping telomere crisis[34] appear to arise independently of NHEJ. It thus remains unclear how DNA lesions from micronuclei are processed and which DSB repair pathway(s) can generate the range of simple and complex rearrangements that are pervasive in cancer genomes.

To determine how specific DSB repair pathways shape the rearrangement landscape of mitotic errors, we leverage a strategy termed CEN-SELECT, which enables the controlled induction of micronuclei containing the Y chromosome harboring a neomycin-resistance (neo^R) marker[7,17,35]. Using this approach, exposure to doxycycline and auxin (DOX/IAA) induces the replacement of the centromeric histone H3 variant CENP-A with a chimeric mutant that functionally inactivates the Y centromere[7]. Following mitotic mis-segregation into micronuclei and chromosome fragmentation, selection for the Y-encoded neo^R marker allows for the isolation of a diverse spectrum of rearrangement types[17]. By generating a series of gene deletions spanning each DSB repair pathway, we identify the canonical NHEJ pathway as the predominant repair mechanism in forming complex rearrangements following chromothripsis. In the absence of NHEJ, chromosome fragments are rarely reassembled by non-NHEJ DSB repair pathways, resulting in persistent DNA damage within the nucleus as MN bodies that trigger cell cycle arrest.

## Results

### NHEJ is the primary DSB repair pathway for fragmented chromosomes from micronuclei

To study the contributions of each DSB repair pathway to chromothripsis, we first generated isogenic DLD-1 knockout (KO) cells in the background of the CEN-SELECT system (Fig. 1a). This was achieved by delivering Cas9 ribonucleoproteins (RNPs) in complex with sgRNAs targeting eight genes spanning multiple DSB repair-related processes, including canonical NHEJ (PRKDC, encoding DNA-PKcs; LIG4, encoding LIG4; NHEJ1, encoding XLF), DNA end protection (TP53BP1, encoding 53BP1), alt-EJ (POLQ, encoding Polθ), SSA (RAD52, encoding RAD52), HR (RAD54L, encoding RAD54), and DNA end resection (NBN, encoding NBS1). In addition to their critical function in a specific DSB repair pathway, these non-essential genes were selected because cells can survive and maintain a mostly diploid karyotype in its absence. Gene KOs were created using either a single sgRNA to induce insertion/deletion mutations or dual sgRNAs to generate frameshift deletions that are discernable by polymerase chain reaction (PCR) from the bulk cell population and single-cell-derived clones (Supplementary Fig. 1a). Twenty-two clones harboring biallelic inactivation of the target gene were confirmed by PCR, Sanger sequencing, and/or immunoblotting (Supplementary Fig. 1b, c).

As expected, LIG4 and XLF KO clones deficient in NHEJ exhibited cellular sensitivity to ionizing radiation (IR) (Supplementary Fig. 1d) and failure to repair IR-induced DSBs, as determined by persistent phosphorylated histone H2AX (γH2AX) foci in the nucleus (Supplementary Fig. 1e). We note that attempts to perturb HR by deleting both copies of BRCA2 were unsuccessful following screening of >100 clones; thus, we moved forward with RAD54 KO clones, which exhibited a modest yet detectable reduction in HR activity using the established DR-GFP assay (Supplementary Fig. 1f). This partial HR defect may be attributed to overlapping functions with RAD54B and/or RAD51AP1[36–38].

Following DOX/IAA treatment, most KO clones generated micronuclei at a frequency comparable to WT cells with the exception of LIG4 and NBS1, which exhibited slightly increased levels of spontaneous and DOX/IAA-induced micronuclei (Supplementary Fig. 2a, b) likely due to elevated genomic instability at baseline. Nonetheless, induction with DOX/IAA resulted in the shattering of the Y chromosome that can be detected on metaphase spreads by DNA fluorescence in situ hybridization (FISH) across all clones (Supplementary Fig. 2c, d).

Following Y chromosome shattering, the reassembly of the neo^R-containing fragment into a stable derivative chromosome confers long-term resistance to G418 selection and produces rearrangements that can be visualized by cytogenetics (Fig. 1b). Cells that cannot maintain the neo^R fragment are thereby rendered sensitive to G418 selection[17]. Among the 22 KO clones generated, cells lacking core NHEJ components (DNA-PKcs, LIG4, XLF) exhibited decreased survival in G418, indicative of their failure to maintain a functional neo^R marker after micronucleation and fragmentation of the Y chromosome (Fig. 1c). Loss of 53BP1, which indirectly promotes NHEJ, similarly resulted in decreased G418 survival (Fig. 1c). We next compared rearrangement frequencies across the surviving fraction of cells by using two DNA paint probes targeting each half of the Y chromosome to visualize a range of Y chromosome-specific rearrangements (Fig. 1b). NHEJ-deficient cells surviving G418 selection showed an overall decrease in rearrangement frequencies regardless of rearrangement type (Fig. 1c). In contrast, loss of Polθ, RAD52, RAD54, or NBS1 had minimal to no effect on both cell survival under G418 selection and rearrangement frequencies following the induction of Y chromosome micronucleation (Fig. 1c). To exclude contributions arising from inter-

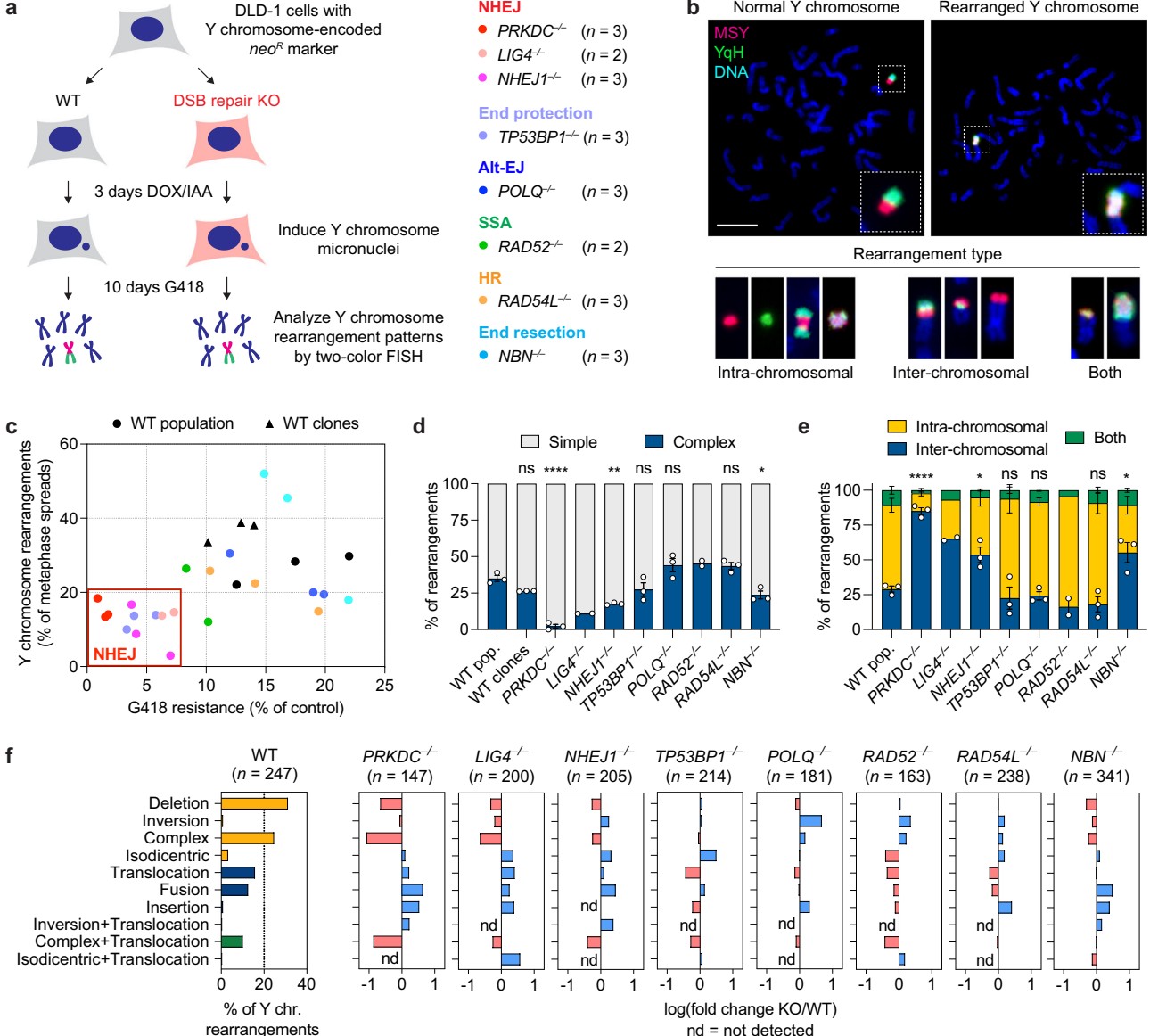

**Fig. 1 | Genomic rearrangement landscape of mis-segregated chromosomes in the absence of specific DNA double-strand break (DSB) repair pathways.**
**a** Experimental approach to survey the impact of specific DSB repair pathways on chromosome rearrangements induced by micronucleus formation. Biallelic gene knockouts (KOs) were generated in the background of the CEN-SELECT system in isogenic DLD-1 cells. Y chromosome-specific mis-segregation into micronuclei and rearrangements were induced by treatment with doxycycline and auxin (DOX/IAA). **b** Representative examples of metaphase spreads with normal or derivative Y chromosomes. Different types of rearrangements can be visualized by DNA fluorescence in situ hybridization (FISH) using probes targeting the euchromatic portion of the male-specific region (MSY, red) and the heterochromatic region (YqH, green) of the Y chromosome. Rearrangements were induced by 3d DOX/IAA treatment followed by G418 selection. Scale bar, 10 μm. **c** Plot summarizing the effect on cell viability after G418 selection (x-axis) and rearrangement frequency of the Y chromosome (y-axis) for each DSB repair KO clone. **d** Proportion of

Y chromosomes exhibiting simple or complex rearrangements, as determined by metaphase FISH, following transient centromere inactivation. **e** Proportion of inter- and/or intra-chromosomal rearrangements. Data in (**d**) and (**e**) represent the mean ± SEM of $n = 3$ independent experiments for WT population, $n = 2$ KO clones for LIG4 and RAD52, and $n = 3$ KO clones for WT, DNA-PKcs, XLF, 53BP1, POLQ, RAD54, and NBS1; statistical analyses were calculated by ordinary one-way ANOVA test with multiple comparisons. In (**d**), ns not significant; ****$p < 0.0001$; **$p = 0.0015$; *$p = 0.0456$. In (**e**), ns not significant; ****$p < 0.0001$; *$p = 0.0229$ ($NHEJ1^{-/-}$) or 0.0142 ($NBN^{-/-}$). **f** Left: distribution of Y chromosome rearrangement types as determined by metaphase FISH following 3d DOX/IAA treatment and G418 selection. Data are pooled from three independent experiments. Right: plots depict the mean fold change in each rearrangement type as compared to WT cells. Sample sizes indicate the number of rearranged Y chromosomes examined; data are pooled from two or three individual KO clones per gene. See also Supplementary Fig. 2. Source data are provided as a Source Data file.

clonal variability, three control clones from the WT population were also isolated and examined, each of which behaved similarly to the parental cells (Fig. 1c).

## Genomic rearrangement landscape of DSB repair deficiency

We next examined how the loss of a specific DSB repair pathway influences the spectrum of rearrangement types generated from

micronuclei formation. In WT cells, the induction of Y chromosome mis-segregation followed by G418 selection for retention of $neo^R$ generated a diverse range of simple and complex intra- and inter-chromosomal rearrangements (Fig. 1d–f, Supplementary Fig. 3), consistent with prior studies[17]. In the absence of NHEJ, however, the rearrangement landscape shifted toward relatively simple inter-chromosomal rearrangements, which were largely comprised of non-

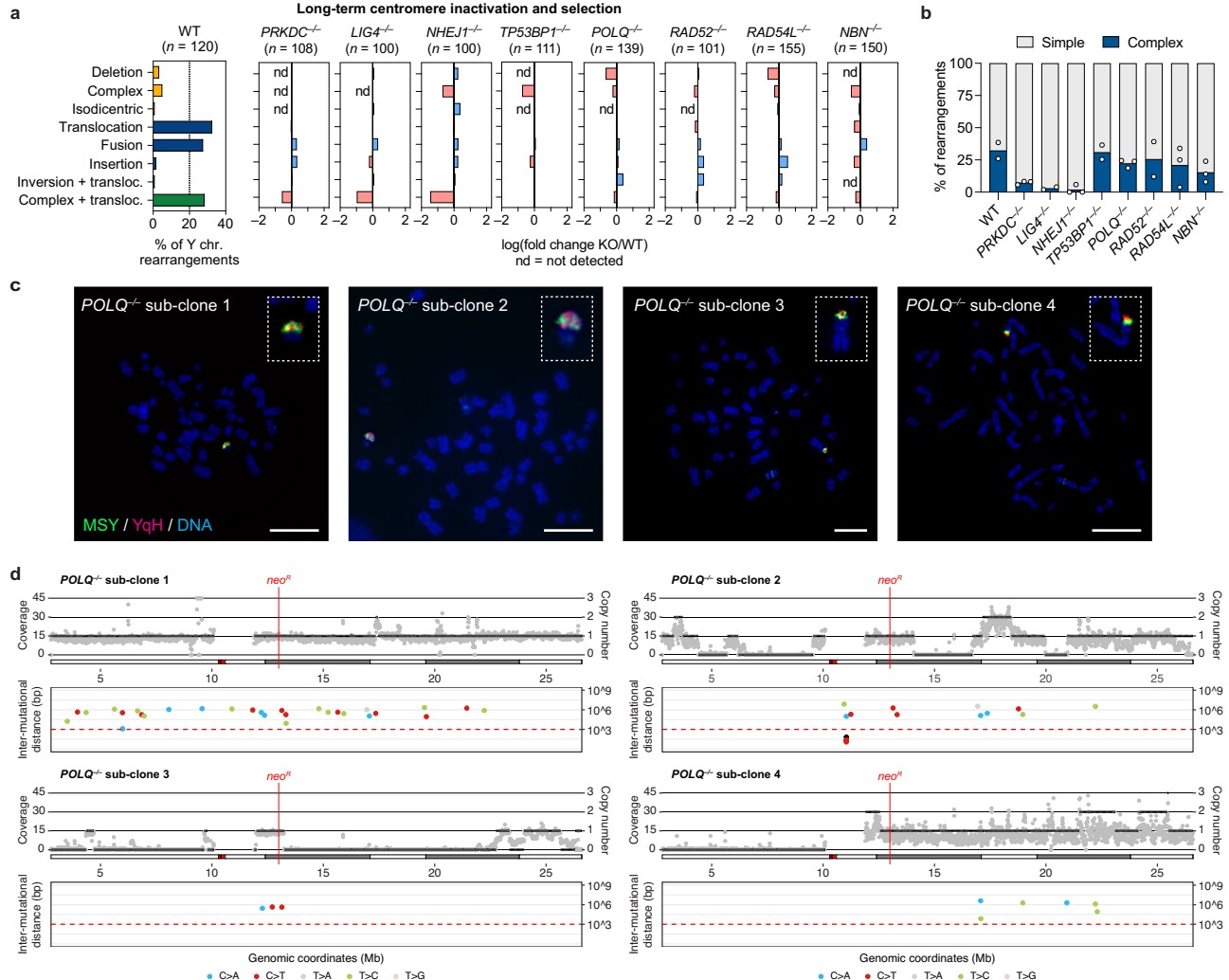

**Fig. 2 | DSB repair pathways beyond NHEJ are dispensable for generating complex rearrangements from micronuclei. a** Left: distribution of Y chromosome rearrangement types as determined by metaphase FISH following continuous passage in DOX/IAA and G418 for ~30 days. Data are pooled from two independent experiments. Right: plots depict the mean fold change in each rearrangement type as compared to WT cells. Sample sizes indicate the number of rearranged Y chromosomes examined; data are pooled from two or three individual KO clones per gene. See also Supplementary Fig. 3. **b** Proportion of Y chromosomes exhibiting simple or complex rearrangements, as determined by metaphase FISH, following sustained centromere inactivation. Data represent the mean of $n = 2$ independent

experiments for WT, $n = 2$ KO clones for LIG4 and RAD52, and $n = 3$ KO clones for DNA-PKcs, XLF, 53BP1, Polθ, RAD54, and NBS1. **c** Cytogenetic characterization of *POLQ* KO sub-clones harboring complex Y chromosome rearrangements following sustained centromere inactivation. Scale bar, 10 μm. **d** Whole-genome sequencing analyses of Polθ KO sub-clones with complex Y chromosome rearrangements exhibiting oscillating DNA copy-number patterns. For each subclone, sequencing depth (gray dots), copy-number information (black lines; top), and inter-mutational distances (bottom) for the mappable regions of the Y chromosome are shown. Source data are provided as a Source Data file.

reciprocal translocations and whole-chromosome fusions (Fig. 1d–f, Supplementary Fig. 3). Notably, there was a sharp reduction in complex rearrangements (Fig. 1d), which were distinguishable by the co-localization of the two-colored FISH probes that are normally separated on metaphase Y chromosomes (Fig. 1b). We previously showed that this cytogenetics-based approach is highly concordant with the use of whole-genome sequencing (WGS) to call chromothripsis events[17]. Reduced intra-chromosomal and complex rearrangements were also observed in cells lacking NBS1, a component of the MRE11–RAD50–NBS1 (MRN) complex, but not 53BP1, Polθ, RAD52, or RAD54 (Fig. 1d–f, Supplementary Fig. 3).

To determine which DSB repair pathways are required to produce stable and genetically heritable derivative chromosomes from micro-nuclei, the panel of KO cells was cultured under sustained centromere inactivation and continuous G418 selection over the span of ~30 days (Supplementary Fig. 4a). In agreement with transient centromere inactivation, WT cells, cells deficient in either the alt-EJ or SSA pathways, or

cells with partial HR defects, exhibited reduced growth rates during early passages followed by a recovery over time (Supplementary Fig. 4b), reflecting cell death owing to the loss of the *neo*[R] marker upon the induction of Y chromosome mis-segregation and the subsequent formation of stable Y chromosome rearrangements[17]. By contrast, NHEJ-deficient cells exhibited a further reduction in growth rate during early passages that was accompanied by a delayed recovery in proliferation (Supplementary Fig. 4b). A similar trend was observed in cells lacking NBS1 (Supplementary Fig. 4b). In agreement with reduced growth in G418, metaphase FISH revealed that NHEJ-deficient cells harbored fewer derivative Y chromosomes, which consisted mostly of simple rearrangements (Fig. 2a, b, Supplementary Fig. 4c).

## NHEJ-deficient cells fail to generate complex rearrangements from micronuclei

A proportion of chromothriptic breakpoint junctions in tumors harbor microhomology[8,14,15], indicative of potential DSB repair by alt-EJ. To

determine whether complex rearrangements were present in alt-EJ-deficient cells, we sequenced four Polθ KO sub-clones that harbored apparent complex rearrangements of the Y chromosome following sustained centromere inactivation, as determined using the previously described two-colored FISH approach (Fig. 2c). Indeed, WGS revealed that three out of four Polθ KO sub-clones exhibited the oscillating DNA copy-number patterns (Fig. 2d) that are characteristic of cancer-associated chromothripsis[8,16]. One subclone harbored a region of clustered C > T hypermutation (Fig. 2d), which has been shown to arise near chromothriptic breakpoints[17,39]. Thus, Polθ-mediated alt-EJ is largely dispensable for the formation of complex rearrangements following chromothripsis from micronuclei.

We next focused on DNA-PKcs-deficient cells for further studies. DNA-PKcs promotes the synapsis of DSB ends by interacting with Ku and activating its kinase activity, which is essential for its function in NHEJ[40–42]. Complementation of DNA-PKcs KO cells with WT DNA-PKcs, but not a K3752R kinase-dead (KD) mutant[40] (Supplementary Fig. 5a), restored the formation of complex Y chromosome rearrangements (Supplementary Fig. 5b–e), confirming that the deficiencies observed in DNA-PKcs KO clones are due to the lack of DNA-PKcs kinase activity. Although rearrangements were markedly reduced in all NHEJ-deficient settings examined, we note that complex rearrangements remained present at a low yet detectable frequency in the absence of NHEJ (Figs. 1d and 2b).

To determine whether DSB repair pathways beyond NHEJ were responsible for the complex rearrangements observed in DNA-PKcs-deficient cells, we inhibited a second DSB repair pathway in DNA-PKcs KO cells using three complementary approaches. First, consistent with our previous findings, small interfering RNA (siRNA)-mediated depletion of components involved in HR (BRCA2), alt-EJ (XRCC1, LIG1, LIG3) and DNA end resection (MRE11) had no effect on either G418 survival rate or the formation of complex rearrangements (Supplementary Fig. 6a–c). Next, we pharmacologically impaired HR by blocking the BRCA2–RAD51 interaction with CAM833[43], alt-EJ with the Polθ inhibitor ART558[44] or the PARP inhibitor olaparib[45], as well as DNA end resection with the MRE11 inhibitor Mirin[46]. Cells lacking DNA-PKcs treated with the indicated inhibitors formed complex rearrangements at a frequency similar to vehicle-treated controls (Supplementary Fig. 6d, e). Lastly, to extend these findings genetically, we used CRISPR/Cas9 editing to generate cells deficient in both DNA-PKcs and a second DSB repair gene (POLQ, RAD52, or RAD54L; Supplementary Fig. 6f). In all double KO settings examined, complex rearrangements remained detectable at a low frequency comparable to loss of DNA-PKcs alone (Supplementary Fig. 6g), suggesting that DSB repair pathways beyond NHEJ are not responsible for the complex rearrangements observed in the absence of DNA-PKcs.

DNA-PKcs promotes DNA end ligation by forming a long-range synaptic complex during NHEJ. However, de novo short-range synaptic complex containing XLF can form in the absence of DNA-PKcs, indicating that DNA end ligation may be possible without DNA-PKcs[47]. This is further suggested by a functional redundancy between DNA-PKcs and XLF in NHEJ[48,49]. To determine whether the residual rearrangements in DNA-PKcs KO cells are formed through minimal NHEJ activity through XLF-mediated end synapsis, we tested for a synergistic reduction in rearrangement formation following depletion of XLF in both WT and DNA-PKcs KO cells. Similar to XLF KO cells, the depletion of XLF in WT cells was sufficient to reduce complex rearrangements (Supplementary Fig. 6h–j). Importantly, complex rearrangements were exceedingly rare in DNA-PKcs KO cells depleted of XLF, occurring in only 3 out of 168 (1.8%) Y chromosomes with rearrangements (Supplementary Fig. 6j). Altogether, these data highlight that DSB repair pathways beyond NHEJ minimally contribute to chromothripsis-induced complex rearrangements.

## Repair kinetics of fragmented chromosomes in MN bodies

Chromosomes encapsulated in micronuclei acquire DSBs, which can be detected by immunostaining for γH2AX. Micronuclei are dysfunctional in sensing and/or repairing DNA damage following rupture of its nuclear envelope[1], suggesting that micronuclear DSBs cannot be repaired until its reincorporation into a functional nucleus[5]. Consistent with this, immunofluorescent staining revealed that micronuclear envelope rupture – as determined by the loss of nucleocytoplasmic compartmentalization[1] or recruitment of the cytosolic DNA sensor cGAS[50,51] – triggered the loss of DNA-PKcs specifically from micronuclei (Supplementary Fig. 7a–d). DNA-PKcs-deficient cells displayed similar levels of γH2AX within micronuclei as compared to WT controls, further supporting that DSBs are not actively repaired within ruptured micronuclei during interphase (Supplementary Fig. 7e, f).

Since NHEJ is suppressed in mitosis[52–54], we hypothesized that most fragments are likely carried over to one or both resulting daughter cell(s) for repair during the subsequent G1 phase. To directly test this, we analyzed DNA damage by immunofluorescence and DNA FISH (IF-FISH) on the previously micronucleated chromosome after its reincorporation into the primary nucleus as an MN body. Compared to WT cells, increased γH2AX and 53BP1 signals were observed on FISH-labeled Y chromosomes manifesting as MN bodies in cells lacking DNA-PKcs and LIG4 (Fig. 3a–d, Supplementary Fig. 7g), indicating that NHEJ engages reincorporated chromosome fragments in the nucleus for repair. Interestingly, loss of NBS1 also resulted in the accumulation of γH2AX (Supplementary Fig. 7g), perhaps due to defects in activating DDR signaling[55–58]. Examination of metaphase spreads for chromosome fragmentation in WT and DNA-PKcs-deficient cells over multiple cell cycles revealed that loss of DNA-PKcs resulted in an accumulation of Y chromosome fragments over time (Fig. 3e, f). Thus, in the absence of NHEJ, fragmented chromosomes reincorporate into the nucleus and persist unrepaired throughout the cell cycle.

We next used live-cell imaging to monitor the kinetics of DSB repair by fusing the minimal focus-forming region (FFR) of 53BP1[59] to a HaloTag (Halo-53BP1). To label and track the Y chromosome from micronuclei into daughter cell nuclei, we used a recently developed dCas9-based SunTag reporter targeting a large repetitive array on the Y chromosome[9]. In the example shown in Fig. 4a, a mother cell with a Y chromosome-specific micronucleus underwent mitosis and subsequently formed a large, Halo-53BP1-labeled MN body in the nucleus of one of the daughter cells. This MN body co-localized with SunTag-labeled Y chromosome fragments, demonstrating that it indeed originated from the micronucleated chromosome from the preceding cell cycle and had now reincorporated into the primary nucleus (Fig. 4a). Whereas the fluorescence intensity of the dCas9-SunTag reporter remained constant throughout the cell cycle, the Halo-53BP1 signal accumulated during early G1 phase, reached a plateau ~10 h after mitosis, and proceeded to gradually decline over a 20-h window during interphase (Fig. 4b, c).

In WT cells, ~23% of micronucleated mother cells formed daughter cells with 53BP1-labeled MN bodies compared to ~3% from non-micronucleated control cells (Fig. 4d, e), confirming that MN bodies are indeed derived from micronuclei in the previous cell cycle. In DMSO-treated control cells, Halo-53BP1 persisted for an average of ~17.9 h until its resolution (Fig. 4f). In the presence of the DNA-PK inhibitor AZD7648[60], MN body-associated Halo-53BP1 signals persisted for the entire duration of imaging, often exceeding 30–40 h (Fig. 4f). We next tracked the fate of daughter cells specifically from micronucleated mother cells. Long-term live-cell imaging revealed that inhibition of DNA-PKcs reduced the proportion of daughter cells that successfully entered another round of mitosis, indicative of cell cycle arrest (Fig. 4g). These findings were further confirmed by depletion of DNA-PKcs (Supplementary Fig. 8a), which similarly exhibited persistent 53BP1 signals (Supplementary Fig. 8b, c) and cell cycle arrest

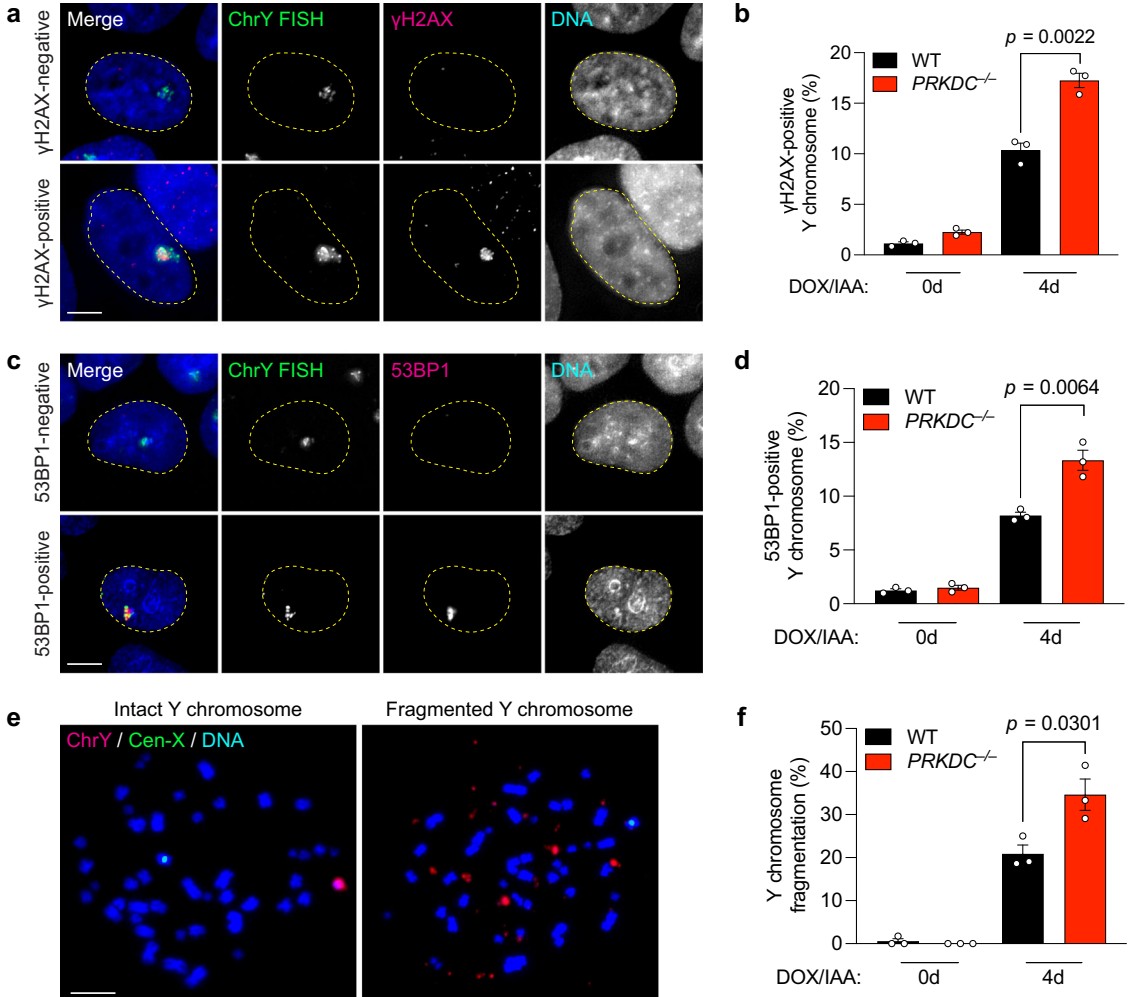

**Fig. 3 | NHEJ-deficient cells accumulate damaged chromosome fragments within MN bodies in the nucleus. a** Images of interphase cells with γH2AX-negative Y chromosome or γH2AX-positive Y chromosomes within an MN body after 4d DOX/IAA treatment. Scale bar, 5 μm. **b** Frequency of Y chromosomes marked by extensive γH2AX in the nucleus. Data were pooled from (left to right): 549, 353, 429, and 293 cells. **c** Images of interphase cells with 53BP1-negative or 53BP1-positive Y chromosomes after 4-day DOX/IAA treatment. Scale bar, 5 μm. **d** Frequency of Y chromosomes marked by extensive 53BP1 in the nucleus. Data were pooled from (left to right): 567, 543, 634, and 592 cells. **e** Images of metaphase spreads with an intact or fragmented Y chromosome after 4d DOX/IAA treatment. Scale bar, 10 μm. **f** Frequency of Y chromosome fragmentation. Data pooled from (left to right): 329, 291, 347, and 373 metaphase spreads. Bar graphs in (**b**), (**d**), and (**f**) represent the mean ± SEM of $n = 3$ independent experiments. Statistical analyses were calculated by two-tailed unpaired Student's $t$-test. Source data are provided as a Source Data file.

(Supplementary Fig. 8d) in the resulting daughter cells. These data suggest that the sustainment of DNA damage signaling as persistent MN bodies can activate the cell cycle checkpoint.

Next, we analyzed γH2AX levels on FISH-labeled fragmented chromosomes that had reincorporated into the nucleus at different time points after mitosis (Fig. 5a). In agreement with live-cell imaging experiments (Fig. 4), most fragmented chromosomes were repaired within 20 h after mitosis in WT cells, whereas DNA-PKcs KO cells continued to harbor γH2AX marks within MN bodies that persisted beyond 20 h (Fig. 5b, c). Since rearrangements are a byproduct of error-prone NHEJ, we sought to directly visualize rearrangements of the Y chromosome in interphase cells at similar time points. To do so, we induced premature chromosome condensation by treatment with the phosphatase inhibitor calyculin A and analyzed metaphase-like chromosomes harboring either unduplicated chromatids from G1-phase cells or sister chromatids from G2-phase cells (Fig. 5d). Calyculin A stimulated efficient chromosome condensation in both G1 and G2-phase cells, as determined by comparing cell cycle profiles by flow cytometry with inspection of metaphase-like spreads (Supplementary Fig. 9a–d). In WT and DNA-PKcs KO cells, most Y chromosomes

remained fragmented 6 h after mitosis during G1 phase (Fig. 5e). As WT cells were allowed to progress throughout the cell cycle into G2 phase, such fragmented chromosomes underwent successful NHEJ to form rearranged chromosomes within 20 h. In contrast, the Y chromosome in DNA-PKcs KO cells remained fragmented, indicative of defects in forming rearrangements in the absence of NHEJ (Fig. 5e). These data provide direct evidence supporting the ligation of reintegrated fragments within MN bodies during a single-cell cycle by the NHEJ pathway.

Lastly, we sought to investigate how inter-chromosomal rearrangements are formed between fragmented micronuclear chromosomes and apparently intact non-homologous chromosomes in the nucleus. To test whether micronuclei-derived chromosome fragments can become improperly ligated onto spontaneous DSBs in the genome, we induced Y chromosome-MN body formation while exposing cells to either low-dose IR or a telomere-incorporating nucleoside analog (6-thio-2′-deoxyguanosine, 6-thio-dG)[61] to induce DNA damage. Both sources of DNA damage were sufficient to elevate the frequency of inter-chromosomal rearrangements involving the Y chromosome (Supplementary Fig. 10a, b). To determine whether chromosome

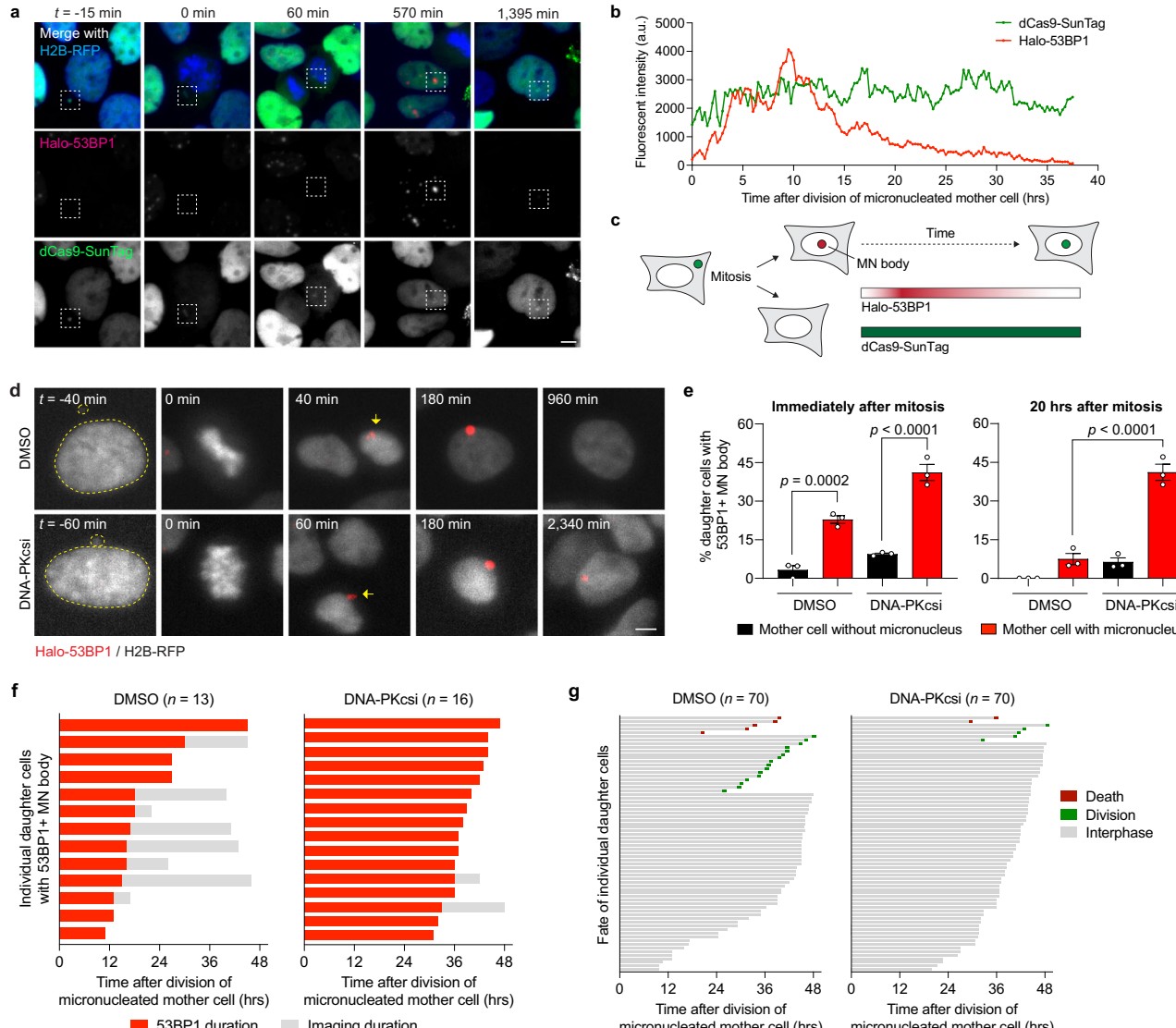

**Fig. 4 | Inhibition of NHEJ prolongs 53BP1 residence time at MN bodies and triggers cell cycle arrest. a** Example time-lapse images of a mother cell harboring a dCas9-SunTag-labeled Y chromosome in a micronucleus undergoing cell division, which in turn generates a daughter cell that incorporates the micronucleated chromosome into the nucleus as an MN body labeled with a HaloTag fused to the minimal focus-forming region of 53BP1 (Halo-53BP1). Time point at 0 min depicts mitotic entry. Scale bar, 5 μm. **b** Measurement of Halo-53BP1 fluorescence intensity over a ~37-h imaging period. The dCas9-SunTag-labeled Y chromosome is monitored as a control. **c** Schematic of Halo-53BP1 residence time at MN bodies from (**a**, **b**) after mitosis. **d** Representative time-lapse images of newly-formed, Halo-53BP1-labeled MN bodies with or without treatment with the DNA-PKcs inhibitor AZD7648. Scale bar, 5 μm. **e** Frequency of Halo-53BP1-labeled MN body formation and persistence. Data represent mean ± SEM of $n = 3$ independent experiments pooled from (left to right): 63, 64, 72, and 74 micronucleated mother cells. Statistical analyses were calculated by ordinary one-way ANOVA test with multiple comparisons. **f** Duration of Halo-53BP1 residence time from (**d**) after MN body formation with or without inhibition of DNA-PKcs in individual daughter cells. **g** Fate of daughter cells after division of micronucleated mother cells. Sample sizes in (**f**, **g**) represent individual daughter cells. Source data are provided as a Source Data file.

fragments from micronuclei can integrate into a targeted genomic locus, we introduced a site-specific DSB by using Cas9 RNPs to trigger the cleavage of two autosomes (chromosomes 3 and 5) or the X chromosome during a window in which MN bodies were present. Indeed, site-specific DSB induction promoted Y chromosome translocations and fusions with the targeted chromosome (Supplementary Fig. 10c, d), indicating that inter-chromosomal rearrangements can be generated by the ligation of chromosome fragments from MN bodies to sites of concurrent DNA damage.

## Discussion

Complex rearrangements arising from chromothripsis are characterized by extensive rearrangements that are confined to one or a few chromosome(s)[8]. Several mutagenic DSB repair pathways can potentially reassemble the fragmented chromosome in seemingly random orientation. Here we demonstrate that fragmented micronuclear chromosomes are predominantly repaired by NHEJ to form complex rearrangements reminiscent of those found in human cancers. Damaged fragments from micronuclei persist throughout mitosis and are carried over into the subsequent cell cycle following its reincorporation into daughter cell nuclei as MN bodies.

Following nuclear reincorporation, analysis of the kinetics of repair suggests that these fragments become reassembled during a prolonged interphase within MN bodies, perhaps due to activation of DNA damage checkpoints and/or the presence of a substantial number of DSB ends that require processing prior to ligation. Live-cell imaging revealed an unexpected delay (by ~10 h) in the onset of 53BP1 recruitment to MN bodies and its subsequent resolution (Fig. 4),

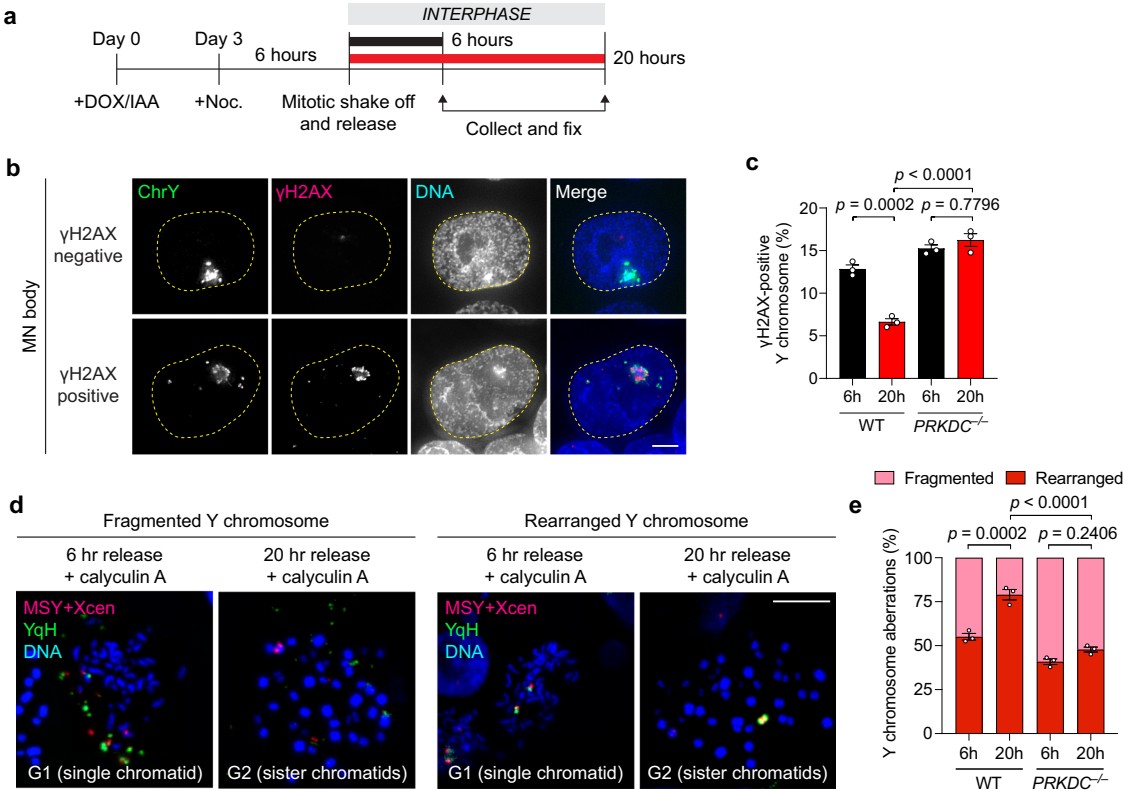

**Fig. 5 | Reincorporated chromosome fragments undergo NHEJ-dependent rearrangement during the ensuing interphase. a** Schematic of experiment to assess DSB repair at 6 and 20 h after mitosis. Noc. nocodazole. **b** Images of interphase cells with γH2AX-negative or γH2AX-positive Y chromosomes before and after division of micronucleated mother cells by IF-FISH. Scale bar, 5 μm. **c** Frequency of γH2AX-positive Y chromosomes in WT and DNA-PKcs KO clone 1 by IF-FISH. Data pooled from (left to right): 352, 361, 341, and 327 Y chromosome-positive interphase cells. **d** Representative images of cells released from

nocodazole arrest for the indicated time points and treated with calyculin A to induce premature chromosome condensation. Chromosome spreads were subjected to DNA FISH using the indicated probes. Scale bar, 10 μm. **e** Frequency of Y chromosome aberrations in WT controls and DNA-PKcs KO clone from (**d**). Data pooled from (left to right): 241, 128, 178, and 159 chromosome spreads. Bar graphs in (**c**) and (**e**) represent mean ± SEM of $n = 3$ independent experiments. Statistical analyses were calculated by ordinary one-way ANOVA test with multiple comparisons. Source data are provided as a Source Data file.

consistent with a prolonged residence time for MDC1 that has been observed on MN bodies[11]. These delays in DSB repair may be caused by the binding of the CIP2A–TOPBP1 complex to the fragmented chromosome during the previous interphase or mitosis[9,10] and that are carried over into the early G1 phase of the cell cycle. Alternatively, ssDNA on micronucleated chromosomes that are acquired during interphase[62] or mitotic entry[63] may require processing upon reincorporation into a functional nucleus to generate fragmented DSB ends that are compatible for ligation by NHEJ.

Analysis of breakpoint junctions from cancer genome sequencing data revealed that most rearrangement events occur without significant microhomology, indicative of ligation between blunt-ended DSB ends. Some junctions exhibit short tracts of microhomology, suggesting a potential contribution by alt-EJ and/or microhomology-mediated break-induced replication[5,14,64,65]. However, our data highlight NHEJ as the predominant, and perhaps exclusive, DSB repair pathway for chromothripsis arising from mitotic errors. Previous studies demonstrated that chromosome fragments from micronuclei can be recognized by the DDR throughout mitosis[9–11], a period in which Polθ-mediated alt-EJ is active[66–68]. Although we cannot fully exclude the possibility that a fraction of fragmented chromosome ends may be repaired by alt-EJ during mitosis, inactivation of the alt-EJ pathway through biallelic deletions in *POLQ* or with small molecule inhibitors targeting Polθ had no measurable impact on the frequency or spectrum of rearrangements produced by micronucleation. We speculate that the binding of the CIP2A–TOPBP1 complex to fragmented chromosomes in micronuclei during the interphase-to-mitosis transition[9]

may preclude engagement by Polθ and/or other alt-EJ components involved in mitosis-specific DSB repair. Indeed, inhibition of the CIP2A–TOPBP1 pathway abolished the formation of MN bodies in interphase cells, which in turn suppressed the formation of complex rearrangements[9].

NHEJ- and NBS1-deficient cells unexpectedly shared several similarities using the assays described here, including delayed growth under sustained centromere inactivation (Supplementary Fig. 4b) and persistent DNA damage within MN bodies (Supplementary Fig. 7g). As part of the MRN complex, NBS1 promotes the resection of DSB ends to generate ssDNA, which diverts DSB repair from NHEJ toward HR and alt-EJ[69,70]. However, the MRN complex can also promote NHEJ through the nuclease activity of MRE11 in processing unligatable ends[71] or by activating the ATM and ATR kinases in response to DSBs and replication stress, respectively[55–58]. Since ATM modulates NHEJ by phosphorylating the C-terminus of DNA-PKcs[72], we speculate that NBS1 promotes complex rearrangements by NHEJ via its regulation of ATM. Similar to LIG4-deficient cells, two NBS1 KO clones also exhibited a higher overall frequency of rearrangements (Fig. 1c), which may reflect elevated baseline levels of genomic instability (Supplementary Fig. 2b). Additionally, 53BP1 deficiency resulted in decreased resistance to G418 and rearrangement frequencies (Fig. 1c), although the rearrangement landscape only partially phenocopied loss of core NHEJ components (Fig. 1d–f, Supplementary Fig. 3). These data suggest that the underlying role of 53BP1 in the repair of fragmented chromosomes may be attributed to additional functions beyond facilitating NHEJ by blocking

end resection, perhaps by promoting mobility and synapsis of DSB ends[73,74] within MN bodies.

In the absence of canonical NHEJ factors, the rearrangements generated from micronuclei formation lack the features of complex rearrangements that are characteristic of chromothripsis. The rearrangement landscape shifts to favor more simple alterations that are typically comprised of unbalanced translocations, whole-chromosome fusions, or chromosome-arm deletions, which could arise from a fraction of micronuclei that harbor relatively few DNA breaks[7] and/or mis-segregated chromosomes that undergo breakage during cytokinesis[75]. These larger chromosome fragments can then ligate to spontaneous DSBs in the genome to generate cytogenetically visible inter-chromosomal rearrangements. By contrast, NHEJ-deficient cells harboring more extensive DNA damage from catastrophically shattered chromosomes that cannot be repaired will ultimately undergo cell cycle arrest. Pharmacological inhibition of NHEJ (e.g., with small molecule inhibitors against DNA-PKcs) may therefore represent a therapeutic avenue to combat chromosomally unstable tumors or those treated with microtubule inhibitors to induce severe mitotic defects. Similar strategies targeting DNA-PKcs may also be effective in suppressing linear chromosome fragments from ligating into a circular extrachromosomal DNAs that can amplify oncogenes and/or genes conferring resistance to anti-cancer therapies[76,77].

## Methods

### Cell lines and reagents
DLD-1 cells were cultured in Dulbecco's Modified Eagle Medium (Thermo Fisher) supplemented with 10% tetracycline-free fetal bovine serum (Omega Scientific) and 100 U/ml penicillin–streptomycin at 37 °C under 5% $CO_2$ atmosphere. Cells were routinely confirmed free of mycoplasma contamination. The derivation of DLD-1 cells expressing the dCas9-SunTag system, mCherry-NLS, and cGAS-GFP were previously described[9]. To generate the 53BP1 reporter system, a HaloTag was fused in-frame to the N-terminus of the minimal FFR (amino acids 1220–1711) of 53BP1 from *TP53BP1* cDNA and cloned into a pBABE-zeo construct (Addgene). DLD-1 cells engineered to carry the dCas9-SunTag system and expressing H2B-mCherry were transduced with retroviruses that were packaged in 293GP cells for 24 h and selected with 50 µg/mL zeocin for 2 weeks. Single-cell-derived clones forming robust 53BP1 foci were isolated and used for live-cell imaging experiments.

Doxycycline (DOX) and auxin (indole-3-acetic acid, IAA) were used at 1 µg/ml and 500 µM, respectively. Nocodazole (Millipore-Sigma) was used at 100 ng/mL for mitotic arrest. Geneticin (G418 Sulfate) was used at 300 mg/mL for selection. Small molecules compounds were used at the following concentrations: 10 µM CAM833 (Tocris Bioscience), 0.5 µM olaparib (Cayman Chemical), 1 µM ART558 (MedChemExpress), 10 µM mirin (MedChemExpress), 1 µM AZD7648 (MedChemExpress), and 0.5 µM 6-thio-dG (a gift from Jerry Shay, UTSW). Dose-response assays were performed to identify an optimal and tolerable drug concentration without affecting DLD-1 cell growth and viability. For rearrangement experiments, inhibitors were added to cells simultaneously with DOX/IAA and incubated for 6 days prior to G418 selection.

### Genome editing
To generate KO clones, TrueCut Cas9 v2 (Thermo Fisher) and sgRNAs (synthesized by Synthego) were assembled into RNP complexes and transfected into cells using Lipofectamine CRISPRMAX Cas9 Transfection Reagent (Thermo Fisher). Seventy-two hours post-transfection, cells were plated at low density (50 cells/10-cm² dish) to isolate single-cell-derived clones. After ~2 weeks, colonies were isolated using cloning cylinders and expanded. Clones were screened by PCR for targeted deletions and confirmed to harbor frameshift mutations by Sanger sequencing. When antibodies were available, immunoblotting was used to confirm the loss of the target protein. All sgRNA sequences and PCR primers used in this study are provided in Supplementary Table 1.

### Cell growth assays
For viability assays, $3 \times 10^4$ cells/well were seeded into six-well plates with or without DOX/IAA treatment. Three days later, cells were washed 3× with PBS and supplemented with fresh media without DOX/IAA. Cells were transferred to 10-cm² plates 3 days later and selected with G418 for 10 days. To calculate relative viability in G418, the total number of cells in the DOX/IAA condition was divided by the total number of cells in the control condition. For quantification of long-term cell growth rates, cells were continuously cultured for ~1 month and the total cell numbers were counted during each passage.

### Cell irradiation and clonogenic assay
Cells were irradiated using a Mark 1 Cesium-137 irradiator (JL Shepherd). For clonogenic assay, cells were plated on six-well plates and irradiated at indicated doses. After 2 weeks, cells were fixed with 100% methanol, incubated with staining solution (0.5% crystal violet in 25% ethanol) for 30 min at room temperature, and washed with water. Colonies were counted and the cell survival rate was normalized to the plating efficiency of untreated control cells.

### HR reporter assay
HR-mediated DSB repair was examined using pDR-GFP reporter assay as previously described with some modifications[78,79]. Briefly, 2 µg pDRGFP was transfected in $1 \times 10^6$ DLD-1 cells along with 2.5 µg pCMV-ISceI and 200 ng DsRed (Clontech) using Lonza Solution V with program T-020. Seventy-two hours later, the cells were trypsinized and resuspended in PBS containing 10% FBS and subsequently examined using a FACSCalibur flow cytometer (BD Biosciences). Data were analyzed using FlowJo (v.10.8.2, BD Biosciences) software.

### Cell cycle profiling
Cells were collected and washed with PBS before fixation with 70% ethanol in PBS for 2 h at −20 °C. Fixed cells were washed twice with PBS followed by incubation with staining solution (100 µg/ml RNase A, 0.1% Triton X-100, 10 µg/ml propidium iodide). After staining, cells were examined using FACSCalibur flow cytometer (BD Biosciences). Cell cycle profiles were generated using FlowJo (v.10.8.2, BD Biosciences) software.

### RNA interference and complementation
Cells were transfected with 20 nM siRNA (Thermo Fisher) using Lipofectamine RNAiMAX (Thermo Fisher) according to the manufacturer's instructions. All siRNA sequences used in this study are provided in Supplementary Table 1. For complementation experiments, a vector containing FLAG-tagged WT or KD DNA-PKcs (K3752R) cDNA was co-transfected with pmaxGFP using a Nucleofector II (Amaxa). Ten days post-transfection, GFP-positive cells were sorted by flow cytometry using a FACSAria (BD Biosciences) into individual wells of a 96-well plate. Clones were expanded and screened by immunoblotting for expression of FLAG-tagged DNA-PKcs.

### Immunoblotting
Cells were collected by trypsinization and pelleted by centrifugation. Cell pellets were washed once with ice-cold PBS and lysed in 2× Laemmli Sample Buffer (50 mM Tris-HCl PH6.8, 2% SDS, 10% glycerol, 0.01% bromophenol blue, 2.5% β-mercaptoethanol). Samples were denatured by boiling at 100 °C for 5 min and resolved by SDS–PAGE. The proteins were transferred to polyvinylidene difluoride membranes using a Trans-Blot Turbo System (Bio-Rad). Blots were blocked with 5% milk in PBST (PBS, 0.1% Tween-20) for 1 h at room temperature before incubation with primary antibodies (1:1000 dilution in PBST except for anti-α-tubulin, which was used at 1:5000) overnight at 4 °C. Blots were

washed 3× in PBST with 10 min each, followed by incubation with horseradish peroxidase-conjugated secondary antibodies (Sigma, 1:5000 dilution in 5% milk in PBST) and an additional three washes in PBST. After adding a chemiluminescent substrate (SuperSignal West Pico PLUS, Thermo Fisher), blots were visualized using a ChemiDoc Imaging System (Bio-Rad). A list of all primary antibodies used in this study is provided in Supplementary Table 2.

## Immunofluorescence

Cells were seeded on chamber slides or coverslips and fixed with 4% formaldehyde for 10 min at room temperature or with ice-cold methanol for 10 min at −20 °C, washed 3× with PBS, and permeabilized with 0.3% Triton X-100 in PBS for 5 min. After washing with PBS, cells were incubated with Triton Block (0.1% Triton X-100, 2.5% FBS, 0.2 M glycine, PBS) for 1 h at room temperature, washed with PBS, and incubated with primary antibodies (1:1000 diluted in Triton Block) overnight at 4 °C, followed by three washes with PBST-X (0.1% Triton X-100 in PBS) 10 min each. After washing, Alexa Fluor-conjugated secondary antibodies (Invitrogen) were diluted 1:1000 in Triton Block and applied to cells for 1 h at room temperature, followed by three washes with PBST-X. Cells were stained with DAPI, rinsed with PBS, air-dried, and mounted with ProLong Gold antifade mounting solution (Invitrogen) before imaging.

## Metaphase spread preparation

To prepare metaphase spreads, cells were treated with 100 ng/ml colcemid (KaryoMAX, Thermo Fisher) for 4 h, collected by trypsinization, and centrifuged at $180 \times g$ for 5 min. Cell pellets were gently resuspended in 500 μL PBS, and 5 mL pre-warmed 0.075 M KCl was added dropwise to the tube while vortexing at low speed. Cells were then incubated at 37 °C for 6 min followed by adding 1 mL of freshly made Carnoy's fixative (3 methanol:1 acetic acid), followed by centrifugation at 180 x $g$ for 5 min and removal of the supernatant. Cell pellets were resuspended in 6 mL ice-cold Carnoy's fixative, followed by centrifugation at 180 x $g$ for 5 min and resuspension in 500 μL Carnoy's fixative. Fixed samples were dropped onto slides and air-dried.

To induce premature chromosome condensation, 100 nM calyculin A (Cell Signaling) was added to directly to the cell culture medium and incubated for 1 h at 37 °C. Cells were then harvested and centrifuged at $180 \times g$ for 5 min. The cell pellets were incubated in 0.075 M KCl followed by fixation in Carnoy's fixative, as described above.

## DNA fluorescence in situ hybridization (FISH)

FISH probes (MetaSystems) were applied to metaphase spreads dropped onto slides. Slides were sealed with a coverslip and denatured at 75 °C for 2 min. After denaturation, samples were incubated at 37 °C overnight in a humidified chamber for hybridization. Samples were then washed with 0.4× SSC at 72 °C for 2 min and 2× SSCT (2× SSC, 0.05% Tween-20) for 30 s at room temperature. The samples were stained with DAPI and mounted with ProLong Gold antifade mounting solution.

For immunofluorescence combined with DNA FISH (IF-FISH), the immunofluorescence procedure was performed first as described above and fixed with Carnoy's fixative for 15 min at room temperature. Samples were rinsed with 80% ethanol and air-dried before proceeding to the FISH protocol.

## Live-cell imaging

To perform live-cell imaging, cells were seeded in 96-well glass-bottom plates (Cellvis, P96-1.5H-N). DLD-1 cells expressing H2B-mCherry, the dCas9-SunTag system, and 53BP1-Halo were treated with DOX/IAA for 72 h, and 200 nM JF646 ligand (Promega) was added 15 min prior to imaging. Images were captured every 20 min for 48 h using an ImageXpress Confocal HT.ai High-Content Imaging System (Molecular Devices) equipped with a 40× objective in a $CO_2$-independent medium (Thermo Fisher) at 37 °C. Images were acquired at $7 \times 1.5$ μm z-sections under low power exposure. Maximum intensity projections were generated using MetaXpress and movies were analyzed using Fiji (v.2.1.0/1.53c).

## Fluorescence microscopy

Metaphase FISH images were obtained using the Metafer Scanning and Imaging Platform (MetaSystems). Briefly, slides were pre-scanned for metaphases using the M-search Mode with a 10× objective (ZEISS Plan-Apochromat 10×/0.45). Image capturing was performed using the Auto-cap Mode with a 63× objective (ZEISS Plan-Apochromat 63×/1.40 oil). Image analyses were performed using Isis Fluorescence Imaging Platform (MetaSystems) and Fiji (v.2.1.0/1.53c).

Immunofluorescent or IF-FISH images were acquired using the DeltaVision Ultra Microscope System (GE Healthcare), which was equipped with a 4.2 MP× sCMOS detector. Images were captured using a 100× objective (UPlanSApo, 1.4 NA) with $15 \times 0.2$-μm z-sections. Images were deconvolved and maximum intensity projections were generated using the softWoRx program (v.7.2.1, Cytiva). Fluorescence intensity was calculated as corrected total cell fluorescence (CTCF) using the formula CTCF = ROI integrated density − (area of the selected ROI × mean background). Images were analyzed using Fiji (v.2.1.0/1.53c).

## Whole-genome sequencing

For WGS, genomic DNA was extracted from ~$3 \times 10^6$ cells by using Quick-DNA Kit (Zymo Research) according to manufacturer's instructions. Sequencing library preparation and WGS were performed by Novogene. Briefly, the genomic DNA of each sample was sheared into short fragments of about 350 bp and ligated with adapters. WGS was performed using a NovaSeq PE150 platform at ~30× coverage.

WGS data were aligned to the GRCh38 [https://www.ncbi.nlm.nih.gov/datasets/genome/GCF_000001405.26] build of the human reference genome using BWA-MEM (v0.7.17)[80]. Aligned sequencing reads were processed using SAMtools (v1.12)[81], and duplicate reads were flagged using Sambamba (v0.8.1)[82]. Sequencing depth was calculated at 10,000 base pair windows using Mosdepth (v0.3.1)[83]. Control-FREEC (v11.6)[84] was used to perform copy-number variation analysis using default parameters. Somatic single-nucleotide mutations were detected using SAGE (v2.8.0, https://github.com/hartwigmedical/hmftools). Visualization of inter-mutation distances across the genome (rainfall plots) was performed using the MutationalPatterns Bioconductor package (v3.10)[85].

## Statistics and reproducibility

Statistical analyses were performed using GraphPad Prism 10.1.1 software using the tests described in the figure legends. For all graphs, $n$ represents the exact sample size as reported in the figure, figure legend, or Source Data file. $P$ values $\leq 0.05$ were considered statistically significant. Statistics were only performed in which $n \geq 3$. All experiments were independently reproduced as described in the figure legends. No statistical method was used to predetermine the sample size. No data were excluded from the analyses. The experiments were not randomized.

## Reporting summary

Further information on research design is available in the Nature Portfolio Reporting Summary linked to this article.

# Data availability

Whole-genome sequencing data presented in this manuscript have been deposited at the European Nucleotide Archive under Project Accession ID PRJEB64431. Source data are provided with this paper.

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

## Acknowledgements

We thank Benjamin Chen and Jerry Shay for providing reagents; Kidist Woldehawariat, Yu-Fen Lin, and Haiyang Yu for assisting with experiments; and members of the Ly Laboratory for discussions. We acknowledge the UT Southwestern Flow Cytometry Core and Peter O'Donnell Jr. Brain Institute for shared use of equipment. This work was supported by the US National Institutes of Health (R35GM146610 and R01CA289435 to P.L.), the Cancer Prevention and Research Institute of Texas (RR180050 to P.L.), and The Welch Foundation (I-2071-20210327 to P.L.). J.E.V.-I. and I.C.-C. acknowledge the European Molecular Biology Laboratory for funding.

## Author contributions

Q.H. and P.L. conceived the project and designed the experiments. Q.H., R.D., A.G., A.M., E.G.M., J.L.E., and H.L. performed experiments and analyzed the data. J.E.V.-I. and I.C.-C. analyzed whole-genome sequencing data. A.J.D., I.C.-C., and P.L. provided supervision. Q.H. and P.L. wrote the manuscript with input from all authors.

## Competing interests

The authors declare no competing interests.

## Additional information

**Supplementary information** The online version contains
supplementary material available at

Peter Ly.

**Peer review information** *Nature Communications* thanks the anon-
ymous reviewers for their contribution to the peer review of this work. A
peer review file is available.

