## [Peer Review File · Nature Communications]

Non-homologous end joining shapes the genomic rearrangement landscape of chromothripsis from mitotic errorsReviewers' Comments:

Reviewer #1:

Remarks to the Author:

This manuscript from Hu et al investigates the DNA repair mechanisms that put back together shattered chromosomes generated following chromosome encapsulation in micronuclei. Micronucleus-associated chromosome shattering is implicated in the process of chromothripsis, a phenomenon of clustered chromosome rearrangements and oscillating copy number patterns. Observations of chromothripsis across many different cancer types make repair of shattered chromosomes an important area of research. The authors approach this question by examining chromosome fragmentation and repair in gene edited cells lacking critical components of NHEJ, end protection, HR and alt-NHEJ. The study is driven by an innovative, previously developed system to conditionally inactivate the Y chromosome centromere and thus micronucleate and fragment the Y chromosome. The ability to anticipate Y chromosome micronucleation allows the authors to query Y chromosome fragmentation and rearrangement/repair using FISH probes and live-cell imaging. This leads the authors to confirm previous findings showing that NHEJ plays a critical role in the repair of chromothripsis chromosomes. Additional experimentation confirms recently published results showing that DNA damage foci can persist on micronucleated chromosomes following reincorporation into the primary nucleus. The approaches used here are innovative, technically rigorous, and generally point towards well-supported conclusions. However, most of the findings presented are in agreement with prior publications from this same group and the findings of others. Therefore, a lack of novelty is the main limitation in this manuscript.

Major point

As the authors point out in the introduction, prior publications from this group have already implicated canonical NHEJ in the ligation of shattered chromosomal fragments (PMID: 27918550). Follow up work from this group using the similar approaches to those described in this manuscript implicated classical NHEJ in micronucleus-associated chromosome rearrangements (PMID: 30833795). Although the methods used here are a technical improvement they largely arrive at the same conclusions of these previous papers. Additionally, other findings in the manuscript including observations of γ H2AX and 53BP1 at 'MN bodies' have previously been reported in the literature (PMID: 37286600). Given the extent of prior reports in this area it is difficult to appreciate the new findings here, which appear to be largely incremental.

Minor points

Fig. 2d: It is difficult to draw conclusions based on the small number of genomes used in this analysis.

Fig. 4g: The authors conclude that DNA-PKcs inhibition causes a cell cycle arrest that blocks entry into mitosis. However, it appears that the great majority of cells are blocked in interphase even in the DMSO conditions. This result would suggest that cell cycle blocks (or technical issues related to prolonged time-lapse imaging) are in play regardless of DNA-PKcs inhibition.

Reviewer #2:

Remarks to the Author:

Hu and colleagues explore how the major DSB repair pathways contribute to chromothripsis using their elegant system to selectively induce mis-segregation of chromosome Y (CEN-SELECT). CEN-SELECT was previously optimized and characterized in the DLD-1 cell line they use here (Ly et al, 2019) and c-NHEJ was implicated as the main DSB repair pathway in that work. Here, Hu and colleagues build upon previous work by 1) employing a CRISPR-Cas9-based gene knock-out (KO) strategy instead of siRNA-mediated knockdown as well as chemical inhibitors in select experiments; 2)

carrying out long-term experiments to track how over time the distribution of genomic rearrangements of the mis-segregated Y chromosome is changing; 3) greatly expanding number of genes and DSB pathways profiled; 4) further exploring the role and dynamics of c-NHEJ repair pathway in generating chromotripsis; and 5) showing that the decreased viability in absence of c-NHEJ is due to cell cycle arrest caused by persistent unrepaired Y chromosome fragments. This is an interesting study worthy of publication in Nat. Comm..

The experiments are well-executed, statistically robust, and mostly well-designed. Note for example the nice use of a FISH-based assay to profile the distribution of simple/complex and intra-/inter-chromosomal rearrangements of a mis-segregated Y chromosome (Figure 1C-F). Another well-designed and executed set of experiments is Halo-53BP1 signal intensity tracking in MN-body cells to determine repair timings using a live-imaging approach (Figure 4A,B), and the usage of calyculin A to assay repair timings in synchronized cells using FISH-based method (Figure 5D,E). However, some of the results do not completely agree with the authors' interpretations and there are a few claims that are not directly backed up by the data.

Main issues:

1. This manuscript aims to definitively describe the DSB repair pathway(s) that lead to chromotripsis upon chromosome mis-segregation. To this end, they test genes required for c-NHEJ, HR, SSA, and alt-EJ. However, their choice of RAD54L for HR is somewhat concerning since prior work reveals only a mild HR defect upon its KO, very likely due to redundancy with RAD54B and RAD51AP1 (Henson et al., 2006; Gottipati et al., 2010; Spies et al., 2016; Selemenakis et al., 2022). The authors cite a (somewhat outdated) review by Hannah Klein who also states: "In mouse ES cells, loss of Rad54 results in a slightly reduced HR frequency, sensitivity to ionizing radiation and mitomycin C, and aberrant repair of DNA damage, whereas *rad54*^{-/-} mice appear to be normal (183, 184). This stands in contrast to other mammalian HR genes such as RAD51, as loss of Rad51 in mouse ES cells is lethal and the mouse *rad51*^{-/-} genotype is an embryonic lethal (6, 185)." Unless the authors can show that HR is severely reduced (DR-GFP assay?) in their RAD54L KO, knockout of BRCA2 might be a better choice to query HR.
2. Similarly, the authors use CAM388, a drug that interferes with BRCA2-Rad51 interaction to query the role of HR in absence of DNA-PKcs. According to the Scott et al. paper they cite, CAM388 has a minimal effect (10-20% reduction) on HR (DR-GFP assay) at the concentration they use (10 μ M). Another method needs to be used to rule out the role of HR in NHEJ deficient cells. They also use RAD54L KO in this part of the study but as detailed above, that approach is also not solid.
3. NBN (Nbs1) is presented as a 'resection' factor. However, Nbs1 has a number of other critical roles, including in activation of ATM signaling and in DNA replication. The text should be revised to reflect this (with references). Since MR (without Nbs1/Xrs2) can mediate some resection (at least in yeast), it may be advisable to confirm the Nbs1 KO data with mirin in at least some of the key experiments.
4. The authors argue (lines 358-361) that c-NHEJ-deficient cells (PRKDC KO) harbor more persistent unrepaired fragments of chromosome Y (Figure 3E-F) which in turn trigger cell cycle arrest. However, the correlation between loss of viability and Y chromosome fragments is not so clear (to this reviewer). For instance, the LIG4 KO clones have fewer metaphases with detectable chromosome Y fragmentation compared to wild-type (WT) cells but are less viable (Extended Figure 1G); the NBN KO clones have similar levels of chromosome Y fragmentation compared to WT (Extended Figure 1G), while having decreased viability (Extended Figure 3B); and the RAD52 KO clones have higher levels of chromosome Y fragmentation than WT cells (Extended Figure 1G), but are equally viable (Extended Figure 3B). Perhaps things could be clarified if they score the % of γ H2AX-positive cells in these knockouts (as in Figure 3B) to see whether there is a lack of sustained DNA damage response when c-NHEJ is intact. Interestingly, it seems that viability correlates well with a greater increase in fusions (Extended Figure 3B,C). Perhaps the authors can comment on this?
5. LIG4 KO might not be complete as it seems there is a faint LIG4 band still present in KO #1 and especially KO #2 (Extended Figure 1C). Can the authors confirm that these cells are c-NHEJ deficient (e.g., determining the rate of γ H2AX foci reduction after radiation in comparison to the XLF KO and 'wt' cells)?

6. Figure 4D-E shows convincingly that DNA-PKcs inhibition leads to persistent DNA damage signaling. However, the claim that DNA damage signaling leads to a cell cycle arrest (p5, line 108; p12, lines 271-273; p16, line 361) is based on 17 WT cells that underwent division (presumably in a field of view of the microscope) vs. 5 cells that underwent division in DNA-PKcs inhibited population. These numbers are very low/insufficient. The authors may want to perform FACS analysis (on a bulk cell population) to address the issue. Also, it would be good to perform DNA-PKcs siRNA knockdown, not just DNA-PKcs inhibition, since effects of DNA-PKcs chemical inhibition and genetic perturbation are quite different.

7. The effect of 53BP1 KO is strong and convincing in Figure 1C and the authors state: "Loss of 53BP1, which promotes NHEJ by protecting DSB ends from undergoing resection, similarly resulted in decreased G418 survival (Figure 1c)." However, according to the literature, 53BP1 has very little effect on general c-NHEJ. For instance, the c-NHEJ mediated repair of DSBs induced by IR is largely unaffected by 53BP1. 53BP1 does affect c-NHEJ in specific settings, such as a repair of distant breaks (class switch and VDJ between distant sites) and telomeres. What they are likely observing here in the repair of Y chromosome fragments (leading to the effects in Figure 1C) may be similar to these settings where DNA ends need to be brought together from distant locations so that 53BP1's ability to promote synapsis and DSB mobility affect the outcome. This interpretation is also consistent with the finding that loss of 53BP1 does not phenocopy loss of genuine c-NHEJ factors (Figure 1D, E). The authors may want to rephrase their interpretation and refer to the relevant papers or a recent review.

Minor issues

1. In all cases, knockouts of relevant genes are studied in clonal cell lines and then compared to the parental line. Since there may be clonal variation in the DLD-1 line, it would be good to pick three control clones and show that clonal variation is minimal for the most important parameters studied.
2. To avoid potential confusion of those readers not fully familiar with the official gene names that differ from the well-known protein names, the authors may consider using phrasing such as XLF-deficient cells rather than NHEJ1 KO cells (or absence of XLF etc.).
3. To show comparable levels of micronuclei and Y chromosome fragmentation between WT cells and KO clones (p6, lines 123-126) one-way ANOVA test should be run between all +DOX/IAA 4 days conditions, not +/- DOX/IAA conditions (Extended Figure 1E, 1G).
4. Figure 1A – Please add timing to the schematic (3 days DOX/IAA, 10 days G418).
5. Figure 1C – The y-axis should say "G418 resistance %" (not sensitivity).
6. Figure 1F – Can the authors remove the "nd" from the graph? That the result is not different is self-evident and some readers might think that nd means not determined.
7. Extended Figure 3A – Please add timing to the schematic (30 days DOX/IAA + G418)
8. 6-thio-dG treatment induces telomere damage but also induces damage elsewhere in the genome (Mender et al., 2015). Thus, it is not possible to say that it is specifically DNA damage at telomeres (p13, line 299) that led to an increase in Y rearrangement with another chromosome upon 6-thio-dG treatment.
9. Line 268: "53BP1 signals persisted during the entire duration of imaging" Delete during (or replace by for).
10. Line 293-296: can the authors rephrase this wordy run-on sentence? For instance: Lastly, we investigated the inter-chromosomal rearrangements between fragmented micronuclear chromosomes and apparently intact chromosomes in the primary nucleus. Delete the second part of the sentence after the comma, which is self-evident (how else?) or add a second sentence: Specifically, we asked whether the MN-derived chromosome fragments were ligated onto DSBs in the rest of the genome.

Reviewer #3:

Remarks to the Author:

The authors study the reassembly mechanisms by which micronuclei with entrapped chromosomes susceptible to catastrophic fragmentation, generated by errors in mitosis, initiate chromothripsis. Chromothripsis generates a spectrum of simple and complex genomic rearrangements that are often

associated with human cancers. They focus on the DSB repair pathways underpinning reassembly. This is important because DSB repair pathway engagement in the genomic reassembly is only incompletely understood. Highly relevant here is the phase of the cell cycle in which chromosome thripsis occurs, as well as the phase in which their genomic re-incorporation will occur, as this determines the spectrum of DSB repair pathways available for the reassembly.

The authors use CRISPR/Cas9 to systematically inactivate distinct DSB processing repair pathways and interrogated the rearrangement landscape of fragmented chromosomes from micronuclei using a model of Y-chromosome micronucleus formation they previously developed and validated. They report that deletion of NHEJ components reduces the formation of complex rearrangements, with the remaining simple alterations failing to show the characteristic patterns of chromothripsis. The authors show that following reincorporation into the nucleus, fragmented chromosomes localize within MN bodies and undergo successful ligation by NHEJ, within a single cell cycle. The authors further report that in the absence of NHEJ, chromosome fragments are rarely engaged by Pol θ -mediated alt-EJ or recombination-based mechanisms, that they show delayed repair, 53BP1-labeled MN bodies and prolonged DDR signaling causing cell cycle arrest. These results are interpreted as evidence that NHEJ is perhaps the exclusive DSB repair pathway generating the complex rearrangements of chromothripsis.

The topic is particularly interesting and the paper covers a spectrum of interesting topics that present an advancement in the field. My only concern is the generality of the conclusion for a key role of NHEJ in chromothripsis – which is actually reduced to the question as to whether generalization of observations with the chromothripsis model employed, which practically excludes significant contributions of other DSB repair pathways, is similarly valid to cancer-associated chromothripsis. Indeed, extensive literature suggests contributions from alt-EJ, although careful analysis of relative contributions, as the authors do, is often lacking

The following specific comments are offered:

1. Whereas alt-EJ often utilizes resection and microhomology, this is not an absolute requirement. This should be considered when analyzing contributions of alt-EJ based on microhomology searches at the generated junctions, which by necessity will underestimate the contribution. Also, Pol θ is responsible for only one subpathway of alt-EJ and PARP1 inhibition frequently has minor effects on DSB processing. Thus, there is room for alt-EJ processing in the model system employed. Perhaps LIG3 and/or LIG1 inhibition may be informative. I recognize that this is a difficult task, but it is important for the paper and its central conclusion on the key role of NHEJ in chromothripsis. Since in the literature, multiple reports suggest the engagement of both c-NHEJ and alt-EJ in chromothripsis, the novelty of the paper is compromised if the exclusion of alt-EJ cannot be made convincing.
2. The authors confine themselves to POLQ knockout/inhibition and PARP1 inhibition to exclude alt-EJ. However, as noted above, it is an established fact that there are multiple alt-EJ pathways and without further evidence, its exclusion a contributor to chromothripsis appears premature.
3. Figure Ext. 1e: There is a factor of 2 increase in micronuclei formation in LIG4 and Nbs1 deficient cells in the absence of DOX/IAA. This may be significant and should be discussed.
4. Fig1c: The increase in Nbs1 deficient cells is of similar magnitude with the decrease following NHEJ deficiency, not minimal, and should be acknowledge and discussed.
5. In general, the aberrant response of NBS1, in almost all endpoints investigated, may be construed as evidence that alt-EJ is involved. It will therefore be essential to diffuse this concern.
6. Authors use the absence of complex chromosomal rearrangements as a proxy to real-life chromothripsis induction. While this may be OK for showing the involvement of main pathways, but may be inadequate and misleading for the multiple alt-EJ pathways, each of which may only have a minor contribution.
7. Ext. Fig. 5g: But the biochemical function of XLF as end-stabilizer may also help alt-EJ.
8. The authors present plenty of convincing evidence that in the absence of NHEJ, chromosome fragments are repaired much slower, which is the expected outcome. However, and again relevant to the conclusions drawn, this fact will require longer periods of observation to underpin the conclusion of minimal alt-EJ contribution to chromothripsis - actually longer than that used in several of the results shown.
9. Fig. 5d: Calyculin A is known to be quite inefficient in inducing premature chromosome

condensation in G1. What the authors analyze may therefore be rare anaphase/G1 cells that retain sensitivity to calyculin A! It is easy to diffuse this concern by showing that the fraction of cells with G1 PCC appearance is similar to the fraction of G1 cells as measured by flow cytometry.

10. It was very nice to see that the authors induced genomic DSBs and examined how they facilitate integration of Y-chromosome fragments. This part strengthens the paper.

We would like to thank the referees for their constructive feedback on our initial submission, which we have carefully considered and addressed in the revised manuscript. We include a point-by-point response to each of the concerns below in red font.

Reviewer #1

This manuscript from Hu et al investigates the DNA repair mechanisms that put back together shattered chromosomes generated following chromosome encapsulation in micronuclei. Micronucleus-associated chromosome shattering is implicated in the process of chromothripsis, a phenomenon of clustered chromosome rearrangements and oscillating copy number patterns. Observations of chromothripsis across many different cancer types make repair of shattered chromosomes an important area of research. The authors approach this question by examining chromosome fragmentation and repair in gene edited cells lacking critical components of NHEJ, end protection, HR and alt-NHEJ. The study is driven by an innovative, previously developed system to conditionally inactivate the Y chromosome centromere and thus micronucleate and fragment the Y chromosome. The ability to anticipate Y chromosome micronucleation allows the authors to query Y chromosome fragmentation and rearrangement/repair using FISH probes and live-cell imaging. This leads the authors to confirm previous findings showing that NHEJ plays a critical role in the repair of chromothripsis chromosomes. Additional experimentation confirms recently published results showing that DNA damage foci can persist on micronucleated chromosomes following reincorporation into the primary nucleus. The approaches used here are innovative, technically rigorous, and generally point towards well-supported conclusions. However, most of the findings presented are in agreement with prior publications from this same group and the findings of others. Therefore, a lack of novelty is the main limitation in this manuscript.

Major point

As the authors point out in the introduction, prior publications from this group have already implicated canonical NHEJ in the ligation of shattered chromosomal fragments (PMID: 27918550). Follow up work from this group using the similar approaches to those described in this manuscript implicated classical NHEJ in micronucleus-associated chromosome rearrangements (PMID: 30833795). Although the methods used here are a technical improvement they largely arrive at the same conclusions of these previous papers. Additionally, other findings in the manuscript including observations of gH2AX and 53BP1 at 'MN bodies' have previously been reported in the literature (PMID: 37286600). Given the extent of prior reports in this area it is difficult to appreciate the new findings here, which appear to be largely incremental.

Thank you for reviewing our manuscript. In regard to novelty, while we previously provided evidence supporting a role for NHEJ in chromothripsis, those experiments had several notable limitations, including the use of RNAi to deplete genes of interest (PMID: 27918550 and 30833795), analysis of only a limited number of genes and pathways (PMID: 27918550 and 30833795), the use of an indirect readout for rearrangements (PMID: 27918550), and/or an exclusive focus on the rearrangement landscape in a DSB repair-proficient WT background (PMID: 30833795).

In the current study, we provide the first comprehensive analysis of genuine DSB repair deficiencies through biallelic deletion of genes spanning multiple pathways in the context of chromothripsis. In addition to providing direct evidence that NHEJ is the major repair pathway in the formation of complex rearrangements following chromothripsis, we further extend the scope of these findings by: 1) demonstrating that non-NHEJ DSB repair pathways do not engage in fragmented chromosomes from micronuclei even in the absence of NHEJ, 2) demonstrating that POLQ-deficient cells continue to harbor the signatures of chromothripsis following the induction of micronuclei, 3) evaluating the kinetics by which NHEJ repair occurs throughout the cell cycle following the reincorporation of the fragmented chromosome into the nucleus, 4) demonstrating that NHEJ-deficient cells accumulate

DNA damage within MN bodies and undergo cell cycle arrest, and 5) exploring mechanisms facilitating the formation of inter-chromosomal rearrangements in the presence of genomic DSBs. Altogether, we believe these findings represent a significant advance in our understanding of the repair mechanisms underlying chromothripsis from micronuclei beyond what has been reported in the literature by us or other groups.

Minor points

Fig. 2d: It is difficult to draw conclusions based on the small number of genomes used in this analysis.

Our whole-genome sequencing analysis was limited to four POLQ-deficient clones because we previously demonstrated that our cytogenetics-based FISH assay can reliably detect complex rearrangements following the induction of chromothripsis (PMID: 30833795; 38359973). This FISH assay leverages two unique paint probes that label each half of the Y chromosome in a distinct color. Following chromosome shattering and the acquisition of complex rearrangements, these two colors merge together and create co-localizing FISH signals that can be visualized by microscopy. In our previous study (PMID: 30833795), we compared ~20 isogenic clones using this FISH approach with whole-genome sequencing. For each clone, we were able to accurately predict chromothripsis by visually inspecting metaphase spreads, which were then confirmed by whole-genome sequencing to harbor all of the characteristic features of chromothripsis in cancer genomes, including random orientation of rearrangement breakpoints and oscillating DNA copy-number patterns.

In the current study, we sought to use sequencing to confirm our FISH data to demonstrate that POLQ-deficient cells continue to generate complex rearrangements. Our FISH-based approach enabled us to interrogate >300 POLQ-deficient cells harboring Y chromosome rearrangements in a highly quantitative manner. Thus, our conclusions are drawn not only from 4 sequenced genomes, but rather from hundreds of single cells.

Fig. 4g: The authors conclude that DNA-PKcs inhibition causes a cell cycle arrest that blocks entry into mitosis. However, it appears that the great majority of cells are blocked in interphase even in the DMSO conditions. This result would suggest that cell cycle blocks (or technical issues related to prolonged time-lapse imaging) are in play regardless of DNA-PKcs inhibition.

We would like to point out that our analyses only include daughter cells that were generated from the division of a mother cell harboring a micronucleus, which explains why many cells do not appear to divide within the 48-hour imaging period in the DMSO conditions. While we cannot fully exclude the possibility of technical issues associated with long-term time-lapse imaging, these data are consistent with a prolonged cell cycle following the formation of MN bodies. We have made these points more explicitly clear in the manuscript text. Additionally, in response to point 6 raised by reviewer #2, we have also repeated this experiment using siRNAs targeting DNA-PKcs and obtained similar results (**Extended Data Figure 8**).

Reviewer #2

Hu and colleagues explore how the major DSB repair pathways contribute to chromothripsis using their elegant system to selectively induce mis-segregation of chromosome Y (CEN-SELECT). CEN-SELECT was previously optimized and characterized in the DLD-1 cell line they use here (Ly et al, 2019) and c-NHEJ was implicated as the main DSB repair pathway in that work. Here, Hu and colleagues build upon previous work by 1) employing a CRISPR-Cas9-based gene knock-out (KO) strategy instead of siRNA-mediated knockdown as well as chemical inhibitors in select experiments; 2) carrying out long-term experiments to track how over time the distribution of genomic rearrangements of the mis-segregated Y chromosome is changing; 3) greatly expanding number of

genes and DSB pathways profiled; 4) further exploring the role and dynamics of c-NHEJ repair pathway in generating chromotripsis; and 5) showing that the decreased viability in absence of c-NHEJ is due to cell cycle arrest caused by persistent unrepaired Y chromosome fragments. This is an interesting study worthy of publication in Nat. Comm..

The experiments are well-executed, statistically robust, and mostly well-designed. Note for example the nice use of a FISH-based assay to profile the distribution of simple/complex and intra-/inter-chromosomal rearrangements of a mis-segregated Y chromosome (Figure 1C-F). Another well-designed and executed set of experiments is Halo-53BP1 signal intensity tracking in MN-body cells to determine repair timings using a live-imaging approach (Figure 4A,B), and the usage of calyculin A to assay repair timings in synchronized cells using FISH-based method (Figure 5D,E). However, some of the results do not completely agree with the authors' interpretations and there are a few claims that are not directly backed up by the data.

We thank the referee for the expert summary of our study, as well as the enthusiasm in the approaches that we implemented.

Main issues:

1. This manuscript aims to definitively describe the DSB repair pathway(s) that lead to chromotripsis upon chromosome mis-segregation. To this end, they test genes required for c-NHEJ, HR, SSA, and alt-EJ. However, their choice of RAD54L for HR is somewhat concerning since prior work reveals only a mild HR defect upon its KO, very likely due to redundancy with RAD54B and RAD51AP1 (Henson et al., 2006; Gottipati et al., 2010; Spies et al., 2016; Selemenakis et al., 2022). The authors cite a (somewhat outdated) review by Hannah Klein who also states: "In mouse ES cells, loss of Rad54 results in a slightly reduced HR frequency, sensitivity to ionizing radiation and mitomycin C, and aberrant repair of DNA damage, whereas *rad54*^{-/-} mice appear to be normal (183, 184). This stands in contrast to other mammalian HR genes such as RAD51, as loss of Rad51 in mouse ES cells is lethal and the mouse *rad51*^{-/-} genotype is an embryonic lethal (6, 185)." Unless the authors can show that HR is severely reduced (DR-GFP assay?) in their RAD54L KO, knockout of BRCA2 might be a better choice to query HR.

As the reviewer suggested, we attempted to generate BRCA2 KO cells by CRISPR-Cas9-mediated gene editing. After a large effort to screen 104 single cell-derived clones, we obtained 20 clones with a heterozygous KO and 10 clones with homozygous deletions that unfortunately did not result in a frameshift mutation, as determined by Sanger sequencing. Thus, we were unable to obtain any homozygous KO clones, a strong indication that complete loss of HR is incompatible with cell viability in the genetic background of DLD-1 cells. We suspect that complete inactivation of p53 may be required to tolerate loss of BRCA2 (as BRCA2 KOs have only been successfully generated in DLD-1 *TP53*^{-/-} cells). To overcome the lethality of BRCA2 loss, we instead depleted BRCA2 by RNA interference and repeated the experiments to analyze rearrangement frequencies. Consistent with a lack of HR involvement in chromotripsis, we did not find any significant differences in rearrangements upon depletion of BRCA2. These new results are included in the manuscript (**Extended Data Figure 6a-c**).

Because of the strong viability defects associated with complete loss of HR, we decided to move forward with RAD54L KOs. As suggested by the reviewer, we tested the efficiency of HR using the DR-GFP assay in our WT and RAD54L KO clones and confirmed that loss of RAD54L resulted in a modest yet statistically significant decrease in measurable HR activity. These data are now included in **Extended Data Figure 1f**. We have also revised the text to mention the possible functional redundancy of RAD54L with RAD54B and RAD51AP1 along with new references.

2. Similarly, the authors use CAM388, a drug that interferes with BRCA2-Rad51 interaction to query the role of HR in absence of DNA-PKcs. According to the Scott et al. paper they cite, CAM388 has a minimal effect (10-20% reduction) on HR (DR-GFP assay) at the concentration they use (10 μ M). Another method needs to be used to rule out the role of HR in NHEJ deficient cells. They also use RAD54L KO in this part of the study but as detailed above, that approach is also not solid.

The 10 μ M dose of CAM833 was selected from six-point dose-response assay that we previously conducted in DLD-1 cells, which was the maximally tolerated concentration without a major impact on cell viability over a six-day period. We have included these data for the reviewer below.

As suggested by the reviewer, we further sought an alternative approach to inhibit HR. To do so, we used siRNAs to deplete BRCA2 in WT and DNA-PKcs-deficient cells and analyzed the resulting rearrangement profiles. Consistent with our previous findings using RAD54L KO or HR-specific inhibitors, we did not observe significant changes in rearrangements compared to controls. These results are included in the revised manuscript (**Extended Data Figure 6a-c**).

3. NBN (Nbs1) is presented as a 'resection' factor. However, Nbs1 has a number of other critical roles, including in activation of ATM signaling and in DNA replication. The text should be revised to reflect this (with references). Since MR (without Nbs1/Xrs2) can mediate some resection (at least in yeast), it may be advisable to confirm the Nbs1 KO data with mirin in at least some of the key experiments.

Thank you for the thoughtful suggestions. We have revised the Discussion section of the manuscript accordingly to reflect the additional critical functions of NBS1 in ATM activation along with citing several appropriate references. To test the requirement of resection-dependent DSB repair, we inhibited MRE11 using Mirin as suggested and also used siRNAs targeting MRE11 in our system. Inhibition or depletion of MRE11 did not significantly influence sensitivity to G418 selection or alter the frequency of generating rearrangements upon the induction of micronuclei. These new results are included in **Extended Data Figure 6a-e**.

4. The authors argue (lines 358-361) that c-NHEJ-deficient cells (PRKDC KO) harbor more persistent unrepaired fragments of chromosome Y (Figure 3E-F) which in turn trigger cell cycle arrest. However, the correlation between loss of viability and Y chromosome fragments is not so clear (to this reviewer). For instance, the LIG4 KO clones have fewer metaphases with detectable chromosome Y fragmentation compared to wild-type (WT) cells but are less viable (Extended Figure 1G); the NBN KO clones have similar levels of chromosome Y fragmentation compared to WT (Extended Figure 1G), while having decreased viability (Extended Figure 3B); and the RAD52 KO clones have higher levels of chromosome Y fragmentation than WT cells (Extended Figure 1G), but are equally viable (Extended Figure 3B). Perhaps things could be clarified if they score the % of γ H2AX-positive cells in these knockouts (as in Figure 3B) to see whether there is a lack of sustained DNA damage response when c-NHEJ is intact. Interestingly, it seems that viability correlates well

with a greater increase in fusions (Extended Figure 3B,C). Perhaps the authors can comment on this?

As suggested, we measured the percentage of γ H2AX-positive MN bodies in LIG4, RAD52 and NBS1 KO cells and found that both loss of LIG4 and NBS1 resulted in an accumulation of DNA damage following reincorporation of the micronucleated chromosomes into daughter cell nuclei. In contrast, RAD52 KO had similar levels of DNA damage compared with WT (**Extended Data Figure 7g**). We further describe a potential function of NBS1 in promoting chromosome rearrangements through activation of ATM in the Discussion (see also response to point 4 by reviewer #3).

In regard to the correlation between cell viability and the increase in translocations/fusions, NHEJ-deficient cells undergoing extensive chromosome fragmentation are sensitized to G418 selection as the chromosome cannot be properly reassembled into a genetically-stable, derivative chromosome. However, cells with less extensive damage (e.g., those with only one or a few DSBs on the micronucleated chromosome) are more likely to survive G418 selection by repairing the broken chromosome to a spontaneous DSB on a different chromosome, in turn generating a translocation/fusion (as demonstrated in **Extended Data Figure 10**).

5. LIG4 KO might not be complete as it seems there is a faint LIG4 band still present in KO #1 and especially KO #2 (Extended Figure 1C). Can the authors confirm that these cells are c-NHEJ deficient (e.g., determining the rate of γ H2AX foci reduction after radiation in comparison to the XLF KO and 'wt' cells)?

Thank you for pointing this out. To confirm the LIG4 KO are c-NHEJ deficient, we conducted the following experiments: First, we performed clonogenic assays following exposure to increasing doses of ionizing radiation (IR) and found that LIG4 KO clones are similarly sensitive to IR as compared to cells lacking XLF (**Extended Data Figure 1d**). Next, as suggested by the reviewer, we irradiated cells with 1 Gy IR and compared the intensity of γ H2AX at different time points before and after IR. Following recovery from IR for 24 hours, whereas WT cells are capable of repairing DNA damage (as determined by a reduction in γ H2AX intensity), clones lacking LIG4 or XLF accumulated persistent γ H2AX foci (**Extended Data Figure 1e**), demonstrating that these KO clones are indeed defective in NHEJ.

6. Figure 4D-E shows convincingly that DNA-PKcs inhibition leads to persistent DNA damage signaling. However, the claim that DNA damage signaling leads to a cell cycle arrest (p5, line 108; p12, lines 271-273; p16, line 361) is based on 17 WT cells that underwent division (presumably in a field of view of the microscope) vs. 5 cells that underwent division in DNA-PKcs inhibited population. These numbers are very low/insufficient. The authors may want to perform FACS analysis (on a bulk cell population) to address the issue. Also, it would good to perform DNA-PKcs siRNA knockdown, not just DNA-PKcs inhibition, since effects of DNA-PKcs chemical inhibition and genetic perturbation are quite different.

In these experiments, we note that only daughter cells that were generated from the division of a mother cell with a micronucleus were analyzed. FACS analysis on a bulk cell population would not be able to discriminate between daughter cells that were derived from micronucleated vs. non-micronucleated mother cells. In regard to sample size, we tracked the signal of 53BP1-Halo for at least 20 hours in more than 120 individual daughter cells that were generated from the division of ~60 micronucleated mother cells (legend of **Figure 4e**). Additionally, the profiles shown in **Figure 4g** each represent the fate of 70 individual daughter cells for both control and DNA-PKcs-inhibited conditions over a 48-hour imaging period. We hope the reviewer can appreciate that these experiments required significant effort and technical expertise to properly execute and analyze. As suggested, we also repeated these live-cell imaging experiments following siRNA-mediated

depletion of DNA-PKcs, which yielded similar results to DNA-PKcs inhibition. These new data are included in the revised manuscript (**Extended Data Figure 8**).

7. The effect of 53BP1 KO is strong and convincing in Figure 1C and the authors state: “Loss of 53BP1, which promotes NHEJ by protecting DSB ends from undergoing resection, similarly resulted in decreased G418 survival (Figure 1c).” However, according to the literature, 53BP1 has very little effect on general c-NHEJ. For instance, the c-NHEJ mediated repair of DSBs induced by IR is largely unaffected by 53BP1. 53BP1 does affect c-NHEJ in specific settings, such a repair of distant breaks (class switch and VDJ between distant sites) and telomeres. What they are likely observing here in the repair of Y chromosome fragments (leading to the effects in Figure 1C) may be similar to these settings where DNA ends need to be brought together from distant locations so that 53BP1’s ability to promote synapsis and DSB mobility affect the outcome. This interpretation is also consistent with the finding that loss of 53BP1 does not phenocopy loss of genuine c-NHEJ factors (Figure 1D, E). The authors may want to rephrase their interpretation and refer to the relevant papers or a recent review.

We thank the referee for this insightful comment regarding 53BP1. We have now modified the statement in the Results section to “Loss of 53BP1, which indirectly promotes NHEJ...”. Additionally, we have added several sentences to the Discussion section describing how loss of 53BP1 only partially phenocopies loss of core NHEJ components and how this may be attributed to its function in synapsis and DSB mobility (along with a citation of a relevant paper and review).

Minor issues

1. In all cases, knockouts of relevant genes are studied in clonal cell lines and then compared to the parental line. Since there may be clonal variation in the DLD-1 line, it would be good to pick three control clones and show that clonal variation is minimal for the most important parameters studied.

The reviewer raises a valid point. To exclude clonal variation, we have now isolated three control WT clones, subjected them to the same DOX/IAA treatment followed by G418 selection procedure, and evaluated their resistance to G418 and rearrangement profiles for each clone. All three WT clones exhibited similar frequencies of G418 resistance and complex rearrangements in a manner comparable to the parental WT population. These data are now included in **Figure 1c-d**.

2. To avoid potential confusion of those readers not fully familiar with the official gene names that differ from the well-known protein names, the authors may consider using phrasing such a XLF-deficient cells rather than NHEJ1 KO cells (or absence of XLF etc.).

To minimize confusion between gene names from their encoded proteins, we added a note for these genes in the first paragraph of the Results section. As suggested, we have also rephrased the text throughout the manuscript to refer to respective genes and their gene products by their more well-known protein names.

3. To show comparable levels of micronuclei and Y chromosome fragmentation between WT cells and KO clones (p6, lines 123-126) one-way ANOVA test should be run between all +DOX/IAA 4 days conditions, not +/- DOX/IAA conditions (Extended Figure 1E, 1G).

As the reviewer suggested, we re-organized these plots and performed a one-way ANOVA test comparing samples following DOX/IAA induction. The conclusions remain unchanged, and these figures have been updated (**Extended Data Figure 2b and 2d**).

4. Figure 1A – Please add timing to the schematic (3 days DOX/IAA, 10 days G418).

We have added timing to the schematic of Figure 1a.

5. Figure 1C – The y-axis should say “G418 resistance %” (not sensitivity).

We have revised the y-axis of Figure 1c.

6. Figure 1F – Can the authors remove the “nd” from the graph? That the result is not different is self-evident and some readers might think that nd means not determined.

We would like to clarify that “nd” in this graph does not refer to “not different”. Instead, it refers to “not detected”. We have made this clearer on the figure by including a new label. We apologize for the confusion.

7. Extended Figure 3A – Please add timing to the schematic (30 days DOX/IAA + G418)

We have updated the schematic of this figure (**Extended Data Figure 4a**) as suggested.

8. 6-thio-dG treatment induces telomere damage but also induces damage elsewhere in the genome (Mender et al., 2015). Thus, it is not possible to say that it is specifically DNA damage at telomeres (p13, line 299) that led to an increase in Y rearrangement with another chromosome upon 6-thio-dG treatment.

We have revised the manuscript to correct this description.

9. Line 268: “53BP1 signals persisted during the entire duration of imaging” Delete during (or replace by for).

We have replaced “during” with “for” in this sentence.

10. Line 293-296: can the authors rephrase this wordy run-on sentence? For instance: Lastly, we investigated the inter-chromosomal rearrangements between fragmented micronuclear chromosomes and apparently intact chromosomes in the primary nucleus. Delete the second part of the sentence after the comma, which is self-evident (how else?) or add a second sentence: Specifically, we asked whether the MN-derived chromosome fragments were ligated onto DSBs in the rest of the genome.

We have rephrased this sentence.

Reviewer #3

The authors study the reassembly mechanisms by which micronuclei with entrapped chromosomes susceptible to catastrophic fragmentation, generated by errors in mitosis, initiate chromothripsis. Chromothripsis generates a spectrum of simple and complex genomic rearrangements that are often associated with human cancers. They focus on the DSB repair pathways underpinning reassembly. This is important because DSB repair pathway engagement in the genomic reassembly is only incompletely understood. Highly relevant here is the phase of the cell cycle in which chromosome thripsis occurs, as well as the phase in which their genomic re-incorporation will occur, as this determines the spectrum of DSB repair pathways available for the reassembly.

The authors use CRISPR/Cas9 to systematically inactivate distinct DSB processing repair pathways and interrogated the rearrangement landscape of fragmented chromosomes from micronuclei using a model of Y-chromosome micronucleus formation they previously developed and validated. They report that deletion of NHEJ components reduces the formation of complex rearrangements, with the

remaining simple alterations failing to show the characteristic patterns of chromothripsis. The authors show that following reincorporation into the nucleus, fragmented chromosomes localize within MN bodies and undergo successful ligation by NHEJ, within a single cell cycle. The authors further report that in the absence of NHEJ, chromosome fragments are rarely engaged by Polθ-mediated alt-EJ or recombination-based mechanisms, that they show delayed repair, 53BP1-labeled MN bodies and prolonged DDR signaling causing cell cycle arrest. These results are interpreted as evidence that NHEJ is perhaps the exclusive DSB repair pathway generating the complex rearrangements of chromothripsis.

The topic is particularly interesting and the paper covers a spectrum of interesting topics that present an advancement in the field. My only concern is the generality of the conclusion for a key role of NHEJ in chromothripsis – which is actually reduced to the question as to whether generalization of observations with the chromothripsis model employed, which practically excludes significant contributions of other DSB repair pathways, is similarly valid to cancer-associated chromothripsis. Indeed, extensive literature suggests contributions from alt-EJ, although careful analysis of relative contributions, as the authors do, is often lacking

The following specific comments are offered:

1. Whereas alt-EJ often utilizes resection and microhomology, this is not an absolute requirement. This should be considered when analyzing contributions of alt-EJ based on microhomology searches at the generated junctions, which by necessity will underestimate the contribution. Also, Polθ is responsible for only one subpathway of alt-EJ and PARP1 inhibition frequently has minor effects on DSB processing. Thus, there is room for alt-EJ processing in the model system employed. Perhaps LIG3 and/or LIG1 inhibition may be informative. I recognize that this is a difficult task, but it is important for the paper and its central conclusion on the key role of NHEJ in chromothripsis. Since in the literature, multiple reports suggest the engagement of both c-NHEJ and alt-EJ in chromothripsis, the novelty of the paper is compromised if the exclusion of alt-EJ cannot be made convincing.

Thank you for reviewing our manuscript. As suggested by the reviewer, we have now further investigated the role of alt-EJ by inhibiting XRCC1, LIG1, LIG3, or both LIG1/LIG3 by RNA interference. Our new data show that depletion of these genes phenocopied POLQ deficiency without significant changes in the G418 survival assays or in the formation of complex rearrangements (**Extended Data Figure 6a-c**). We also performed these experiments in DNA-PKcs KO cells and obtained similar results, further supporting a lack of involvement of alt-EJ in reassembling fragmented chromosomes from micronuclei.

2. The authors confine themselves to POLQ knockout/inhibition and PARP1 inhibition to exclude alt-EJ. However, as noted above, it is an established fact that there are multiple alt-EJ pathways and without further evidence, its exclusion a contributor to chromothripsis appears premature.

We completely agree that POLQ-mediated MMEJ represents one sub-pathway of alt-EJ. To further explore the role of alt-EJ, we used siRNAs to deplete additional factors required for alt-EJ, including XRCC1, LIG1, and/or LIG3, as described in the previous response (**Extended Data Figure 6a-c**). While we agree that multiple alt-EJ sub-pathways exist, the mechanisms underlying those beyond POLQ-mediated MMEJ have not been well characterized, thereby making it difficult to systematically evaluate.

3. Figure Ext. 1e: There is a factor of 2 increase in micronuclei formation in LIG4 and Nbs1 deficient cells in the absence of DOX/IAA. This may be significant and should be discussed.

We suspect that the increase in micronuclei in LIG4 and NBS1 KO cells is due to an elevated baseline level of genomic instability. Consistent with this, prior studies have demonstrated that LIG4-

deficient MEFs harbor a higher percentage of structural chromosomal abnormalities compared to loss of DNA-PKcs (PMID: 10823907). In cells lacking NBS1, we suspect that these baseline genomic instability defects may arise from the failure to activate the ATM kinase in response to DSBs. We have included a discussion of this point in the revised manuscript, as well as re-plotted these data to make this comparison clearer (**Extended Data Figure 2b**). That being said, we also note that LIG4 and NBS1 KO cells undergo highly specific Y chromosome shattering upon DOX/IAA induction similarly to WT cells (**Extended Data Figure 2d**).

4. Fig1c: The increase in Nbs1 deficient cells is of similar magnitude with the decrease following NHEJ deficiency, not minimal, and should be acknowledged and discussed.

We have revised the Discussion section to acknowledge this point. Two NBS1-deficient clones indeed generated a higher frequency of Y chromosome rearrangements, although the survival rate of these cells under G418 selection is similar to WT controls. We suspect the increased rearrangements arose from an elevated baseline level of genomic instability rather than from the induction of micronuclei (as discussed above).

5. In general, the aberrant response of NBS1, in almost all endpoints investigated, may be construed as evidence that alt-EJ is involved. It will therefore be essential to diffuse this concern.

Although NBS1 (as part of the MRN complex) indeed participates in DNA end-resection to promote HR or alt-EJ, it is also involved in the general DNA damage response by stimulating the kinase activity of ATM (PMID: 15064416; 15965469; 17486112). NBS1 also participates in facilitating NHEJ during G1 (PMID: 21087997), which represents the phase of the cell cycle when fragmented chromosomes from micronuclei are repaired within MN bodies. We have updated the manuscript to include a discussion of this point along with citing the appropriate references mentioned above.

6. Authors use the absence of complex chromosomal rearrangements as a proxy to real-life chromothripsis induction. While this may be OK for showing the involvement of main pathways, but may be inadequate and misleading for the multiple alt-EJ pathways, each of which may only have a minor contribution.

From our knowledge of the literature, LIG1 and LIG3 is important for alt-EJ. Consistent with POLQ-deficient cells, our new data using siRNAs to deplete LIG1 and LIG3 suggest that alt-EJ has a minimal role in the formation of complex rearrangements from micronuclei (**Extended Data Figure 6a-c**). We have revised the manuscript accordingly.

7. Ext. Fig. 5g: But the biochemical function of XLF as end-stabilizer may also help alt-EJ.

Although we cannot rule out this possibility, we are unaware of any evidence in the literature that implicates an involvement of XLF in alt-EJ. It is generally well accepted that XLF exclusively participates in NHEJ with perhaps a functional redundancy with DNA-PKcs (PMID: 35760797; 23345432). Therefore, we have interpreted our data with the function of XLF in NHEJ instead of alt-EJ in mind.

8. The authors present plenty of convincing evidence that in the absence of NHEJ, chromosome fragments are repaired much slower, which is the expected outcome. However, and again relevant to the conclusions drawn, this fact will require longer periods of observation to underpin the conclusion of minimal alt-EJ contribution to chromothripsis - actually longer than that used in several of the results shown.

We believe that we have reached our technical limits by performing live-cell imaging to track cell fate and 53BP1 signals for over a 48-hour period. While it may be possible to extend such an analysis

further in experiments involving fixed cells, the results from our comprehensive panel of short- and long-term assays have reproducibly demonstrated that POLQ-deficiency, POLQ inhibition, or LIG1/LIG3 depletion has no measurable phenotype in this setting and closely mirrors that of controls. We therefore believe we do not have strong rationale to continue pursuing a role of alt-EJ in the context of chromothripsis using our system.

9. Fig. 5d: Calyculin A is known to be quite inefficient in inducing premature chromosome condensation in G1. What the authors analyze may therefore be rare anaphase/G1 cells that retain sensitivity to calyculin A! It is easy to diffuse this concern by showing that the fraction of cells with G1 PCC appearance is similar to the fraction of G1 cells as measured by flow cytometry.

This is a great suggestion. As the reviewer suggested, we have now compared the fraction of cells in G1, S, or G2 by flow cytometry to an analysis of calyculin A-induced chromosome spreads. These experiments confirmed that the fraction of cells in each cell cycle stage as determined by flow cytometry matches the analysis of chromosome spreads. We have included these results in the revised manuscript (**Extended Data Figure 9**).

10. It was very nice to see that the authors induced genomic DSBs and examined how they facilitate integration of Y-chromosome fragments. This part strengthens the paper.

We thank the reviewer for his/her enthusiasm for these experiments.

Reviewers' Comments:

Reviewer #1:

Remarks to the Author:

The authors have satisfactorily addressed my prior concerns.

Reviewer #2:

Remarks to the Author:

The authors have addressed my concerns. However, since the issue of HR was not fully addressed, I request that where they refer to the effect of HR, they do not use 'inactivate' or similar. They should describe their attempts more accurately as in: partial knockdown and partial impairment.

Reviewer #3:

Remarks to the Author:

The authors have carefully considered comments and suggestions and have provided convincing responses.

We are grateful to the reviewers for their time and constructive feedback on our manuscript.

Reviewer #1

The authors have satisfactorily addressed my prior concerns.

Reviewer #2

The authors have addressed my concerns. However, since the issue of HR was not fully addressed, I request that where they refer to the effect of HR, they do not use 'inactivate' or similar. They should describe their attempts more accurately as in: partial knockdown and partial impairment.

We have revised the manuscript as suggested to reflect a partial inactivation and impairment of HR.

Reviewer #3

The authors have carefully considered comments and suggestions and have provided convincing responses.